# Boundary Guided Learning-Free Semantic Control with Diffusion Models

**Ye Zhu[1,2], Yu Wu[3], Zhiwei Deng[4], Olga Russakovsky[2], Yan Yan[1]**
[1]Department of Computer Science, Illinois Institute of Technology
[2]Department of Computer Science, Princeton University
[3]School of Computer Science, Wuhan University
[4]Google Research
yezhu@princeton.edu, wuyucs@whu.edu.cn, zhiweideng@google.com,
olgarus@princeton.edu, yyan34@iit.edu

## Abstract

Applying pre-trained generative denoising diffusion models (DDMs) for downstream tasks such as image semantic editing usually requires either fine-tuning DDMs or learning auxiliary editing networks in the existing literature. In this work, we present our *BoundaryDiffusion* method for efficient, effective and lightweight semantic control with frozen pre-trained DDMs, without learning any extra networks. As one of the first learning-free diffusion editing works, we start by seeking a comprehensive understanding of the intermediate high-dimensional latent spaces by theoretically and empirically analyzing their probabilistic and geometric behaviors in the Markov chain. We then propose to further explore the critical step for editing in the denoising trajectory that characterizes the convergence of a pre-trained DDM and introduce an automatic search method. Last but not least, in contrast to the conventional understanding that DDMs have relatively poor semantic behaviors, we prove that the critical latent space we found already exhibits semantic subspace boundaries at the generic level in *unconditional* DDMs, which allows us to do controllable manipulation by guiding the denoising trajectory towards the targeted boundary via a *single-step* operation. We conduct extensive experiments on multiple DPMs architectures (DDPM, iDDPM) and datasets (CelebA, CelebA-HQ, LSUN-church, LSUN-bedroom, AFHQ-dog) with different resolutions (64, 256), achieving superior or state-of-the-art performance in various task scenarios (image semantic editing, text-based editing, unconditional semantic control) to demonstrate the effectiveness. Project page at *https://l-yezhu.github.io/BoundaryDiffusion/*.

## 1 Introduction

The denoising diffusion models (DDMs) [53] have been successfully applied in various tasks such as image and video synthesis [21, 41, 22, 48, 14, 23, 52, 20], audio generation [33, 40, 37, 71], image customization [50, 64], reinforcement learning [26, 11], and recently in scientific applications [63, 34, 58]. Among various applications of DDMs, one popular downstream task is to use pre-trained *unconditional* DDMs for image manipulation and editing, by either re-training the diffusion models [31], or learning extra auxiliary neural networks for the editing signals [44, 35]. However, as these methods require additional learning processes, we argue that they have not yet fully leveraged the great potential of pre-trained DDMs. In this work, we show that the latent spaces of pre-trained DDMs [1], when leveraged properly, can be directly used to perform various manipulation tasks without

---

[1]The latent spaces in our context are different from the concept of LDMs (a.k.a., StableDiffudion [48]). While the main idea of LDMs is to operate the diffusion denoising process on the latent space of raw data, we investigate the latent spaces represented by the intermediate diffusion states along the Markov chain.

37th Conference on Neural Information Processing Systems (NeurIPS 2023).

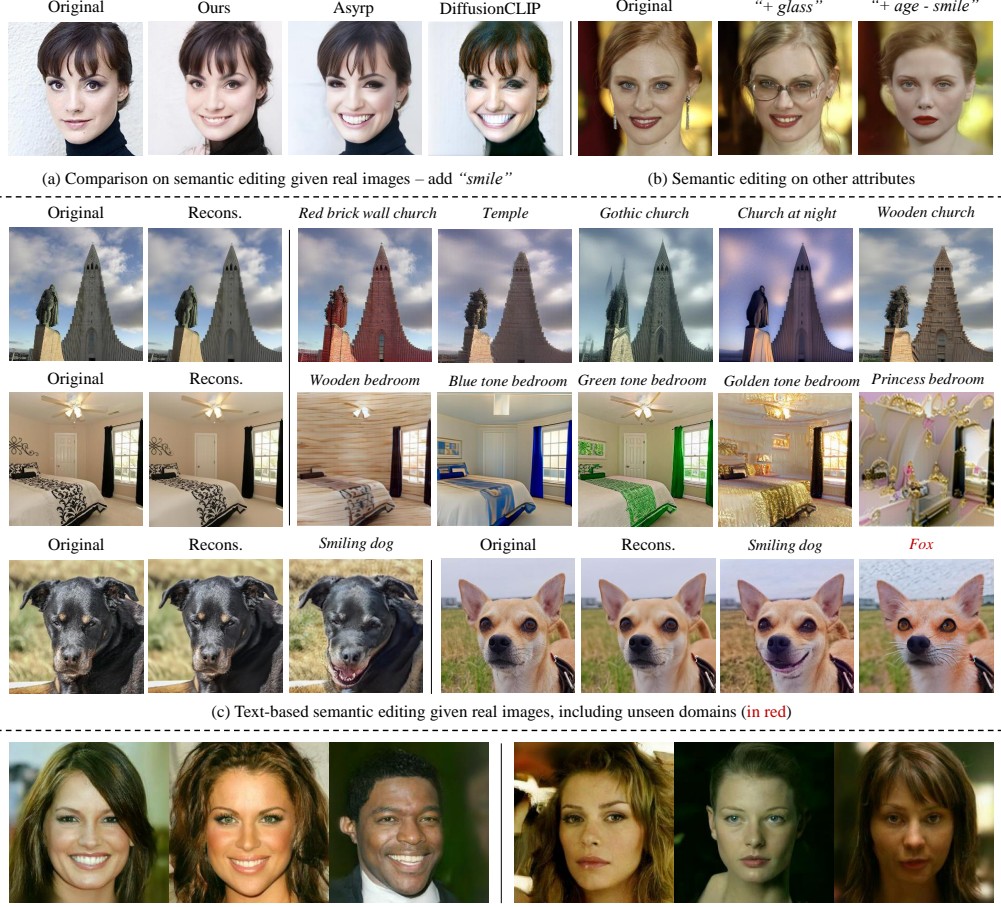

(a) Comparison on semantic editing given real images – add *"smile"*  (b) Semantic editing on other attributes

(c) Text-based semantic editing given real images, including unseen domains (in red)

(d) Unconditional semantic control from sampled latent encodings on the attribute *smile*

Figure 1: We present the *BoundaryDiffusion* for efficient (**single-step**), effective and light-weight semantic control on multiple application scenarios with *frozen* pre-trained DDMs. Different from existing works, our *BoundaryDiffusion* is a **learning-free** method that achieves SOTA performance without learning any extra networks, in contrast to *e.g.* Asyrp [35], which requires training auxiliary editing neural networks, and DiffusionCLIP [31], which fine-tunes the pre-trained DDMs. We include more **randomly selected non-cherry-picked results** in Appendix G as qualitative demonstrations.

learning any extra networks and achieve superior or SOTA performance as shown in Fig. 1. This makes our work the first to achieve learning-free diffusion editing with *unconditional* base models that are trained without additional semantic supervision or conditioning.

The understanding of latent spaces has been a challenging yet critical factor for better explaining, interpreting, and utilizing diffusion models. On the one hand, the DDMs are formulated as a long Markov chain (usually with 1,000 diffusion steps) [21], which introduces the same number of generic latent spaces within a single model. In contrast to a single unified latent space in GANs [16], this unique formulation imposes extra difficulties for studying the latent spaces of DDMs, leading to additional research questions such as *"which latent spaces should we focus on to analyze their behaviors?"* On the other hand, abundant methodology designs for multiple downstream applications have been motivated by their corresponding understanding in terms of latent spaces [44, 31, 35]. For example, DiffAutoencoder [44] proposes to learn auxiliary encoders based on existing latent variables to obtain semantically meaningful representations for image attribute manipulations, mainly because the generic latent spaces in DDMs are initially believed to lack semantics. For the same reason, DiffusionCLIP [31] proposes to achieve semantic manipulation by fine-tuning the entire pre-trained DDMs each time given a new editing target attribute. More recently, Asyrp [35] observes that DDMs already form semantic spaces, but only in the bottleneck level of U-Net implementations [49],

referred to as $h$-space. Accordingly, they propose to learn an auxiliary editing network that operates on the $h$-space for each manipulation attribute. To this end, in this work, we feature our first branch of contributions in terms of latent spaces understanding from the following two aspects, by supplementing and correcting the current literature: **i)** Through the geometric and probabilistic analysis of the latent encodings from the departure space at diffusion step $T$, we demonstrate that the stochastic denoising and deterministic inversion (the process to convert image to noisy latent variables, which is essential for semantic image editing [31, 35]) trajectories along the Markov chain are **asymmetric**, as opposed to previous understanding [35] (see Sec. 3). **ii)** We further propose a theoretically supported automatic search method to identify the critical diffusion step (*i.e.*, the mixing step) along the chain, which starts to exhibit semantics at the *generic* level (referred as $\epsilon$-space), as supplementary and/or correction to existing literature [44, 31, 35] (see Sec. 4 and Sec. 5).

Based on the better understanding of latent spaces, we then propose our *BoundaryDiffusion* method to achieve semantic control with pre-trained frozen and unconditional diffusion models in a learning-free manner in Sec. 5. Specifically, our approach first locates the semantic boundaries in the form of hyperplanes via SVMs within the latent space at the critical mixing step. We then introduce a mixing trajectory with controllable editing strength, which guides the original latent encoding to cross the semantic boundary at the same diffusion step to achieve manipulation given a target editing attribute as in Fig. 3. We also optimize the image quality via a combination of deterministic and stochastic denoising processes. We conduct extensive experiments using different DDMs architectures (DDPM [21], iDDPM [41]) and datasets (CelabA [39], CelebA-HQ [27], AFHQ-dog [10], LSUN-church [66], LSUN-bedroom [66]) for multiple applications (image semantic editing, text-based editing, unconditional semantic control), achieving either superior or SOTA performance in both quantitative scores ($S_{dir}$ [45], segmentation consistency[65, 68, 67], face identity similarity [13] and FID [19]) and qualitative user study evaluations, as presented in Sec. 6. Compared to previous learning-based methods [31, 35], our *BoundaryDiffusion* method features a light-weight and unified single-step operation without any additional training cost. In addition, different from DiffusionCLIP [31] and Astrp [35], which take about 30 min of learning time *per attribute*, our *Boundary* takes approximately negligible time about 1s and as few as 100 images to locate and identify the semantic boundaries.

Overall, our work contributes to both latent space understanding and technical methodology designs. Firstly, we provide multiple novel findings and analysis that correct or supplement the current understanding of latent spaces of DDMs. Notably, we reveal that the previous symmetric assumption of stochastic denoising and deterministic inversion directions does not hold after explicitly analyzing the geometric and probabilistic properties in the departure latent space at diffusion step $T$. We also demonstrate that the unconditionally trained DDMs exhibit meaningful semantics at our located mixing step along the Markov chain in generic $\epsilon$-space level. Secondly, our proposed *BoundaryDiffusion* method achieves semantic control in a learning-free manner, providing a promising direction for resources friendly and efficient downstream applications. The above contributions are further supported and validated by strong performance through extensive experiments and comprehensive evaluations. Code is available at https://github.com/L-YeZhu/BoundaryDiffusion.

## 2    Background

We briefly describe essential background here, and include more detailed related work in Appendix A.

**Denoising Diffusion Models.** Denoising Diffusion Probabilistic Models (DDPMs) [21] are one of the principal formulations for diffusion generative models [53]. The core design of DDPMs consists of a stochastic Markov chain in two directions. The forward process adds stochastic noises (usually parameterized with Gaussian kernel) to a data sample $x_0$ following:

$$q(\mathbf{x}_t|\mathbf{x}_{t-1}) = \mathcal{N}(\sqrt{1-\beta_t}\mathbf{x}_{t-1}, \beta_t\mathbf{I}_d), \tag{1}$$

where $\{\beta_t\}_t^T = 1$ are usually scheduled variance.

The reverse direction, which corresponds to the generative process, denoises a latent sample $\mathbf{x}_T$ (often sampled from a standard Gaussian distribution) to a data sample $\mathbf{x}_0$ as:

$$\mathbf{x}_{t-1} = \frac{1}{\sqrt{1-\beta_t}}(\mathbf{x}_t - \frac{\beta_t}{\sqrt{1-\alpha_t}}\epsilon_t^\theta(\mathbf{x}_t)) + \sigma_t z_t, \tag{2}$$

where $\epsilon_t^\theta$ is the learnable noise predictor, $z_t \sim \mathcal{N}(0, \mathbf{I})$, and the variance of the reverse process $\sigma_t^2$ is set to be $\sigma^2 = \beta$.

Alternatively, denoising diffusion implicit models (DDIMs) consider a non-Markovian process with the same forward marginals as DDPMs, with a slightly modified deterministic *sampling* process as follows:

$$\mathbf{x}_{t-1} = \sqrt{\alpha_{t-1}}(\frac{\mathbf{x}_t - \sqrt{1-\alpha_t}\epsilon_t^\theta(\mathbf{x}_t)}{\sqrt{\alpha_t}}) + \sqrt{1-\alpha_{t-1}-\sigma_t^2} \cdot \epsilon_t^\theta(\mathbf{x}_t) + \sigma_t z_t, \qquad (3)$$

where $\sigma_t = \eta\sqrt{(1-\alpha_{t-1})/(1-\alpha_t)}\sqrt{1-\alpha_t/\alpha_{t-1}}$, as the $\eta$ control the degree of stochasticity. Specifically, the first term indicates the directly predicted $\mathbf{x}_0$ at arbitrary diffusion step $t$, and the second term denotes the direction pointing to $\mathbf{x}_t$, and the last term is the random noise.

In this work, we focus on the study of denoising diffusion models. While the pre-trained models are learned based on DDPMs, we adopt DDIMs for inverting real images $\mathbf{x}_0 \in \mathcal{X}$ to obtain their corresponding latent encodings $\mathbf{x}_t \in \epsilon_t$ as in [31, 35]. However, different from conventional understanding, we reveal this inversion process is **asymmetric** to the stochastic denoising process from DDPMs in Sec. 3, which is an important fact to understand the reason why the previous works require extra learning to achieve the semantic editing effects.

**Semantic Understanding and Trajectory in High Dimensional Space.** Previous works on the semantic understanding of latent space for generative models have largely focused on variants of GANs [16], such as the $\epsilon$ space for generic GANs [16], $\mathcal{P}$ space from ProgressiveGAN [27], and $\mathcal{W}$ space from StyleGAN [29]. More recently, [35] study the latent spaces of diffusion models and show that the pre-trained DDMs already have a semantic latent space at the bottleneck level of U-Net, and name it $h$-space. We observe that there exist two unique components for DDMs compared to previous studies for GANs [51, 12]: the relatively high-dimensionality of intermediate latent spaces and the trajectory along the Markov chain.

Firstly, unlike GANs that usually sample from a lower dimensional Gaussian distribution and upsample the latent encoding to the higher data space [16], DDMs [53] maintain the same dimensionality for all the intermediate latent encodings $\mathbf{x}_{\{1:T\}}$ throughout the entire Markov chain. The above formulation regularizes the latent encodings to stay in higher dimensional spaces, which motivates us to tackle the problem using existing mathematical tools from high dimensional space theory [8], such as the concentration mass distribution and Gaussian radius estimation. In addition, DDMs model a random walk on the data space as a Markov stochastic process [56, 57, 32, 55, 24, 60], and form a trajectory as the diffusion step proceeds. This trajectory characterizes the property of the considered Markov and imposes nature discussion on the convergence question, which inspires us to draw the connection between the mixing time study in Markov chain [38] and diffusion models.

## 3 High-Dimensional Latent Spaces in Diffusion Models

**Notations.** Given a pre-trained DDM $p$ (we omit $\theta$ from common notation $p_\theta$ given the fact that the parameters are frozen in this work) with $T$ diffusion steps. We define the raw data space as $\mathcal{X}$ and a data sample $\mathbf{x}_0 \in \mathcal{X}$ in $d$-dimension. The intermediate latent variables $\mathbf{x}_{\{1:T\}}$ are samples in the same dimensionality as $\mathbf{x}_0$ from their corresponding spaces $\epsilon_{\{1:T\}}$ . Specifically, as previous works suggest there are two methods to obtain the latent encodings, either from direct sampling as most generative works do [21, 14, 48, 33], or from a given raw data $\mathbf{x}_0$ after inversion via DDIMs [31, 35]. We note $\mathbf{x}_t^s$ and $\mathbf{x}_t^i$ as the latent encodings at the $t$-th step from $\epsilon_t$ via sampling and inversion, respectively. Similarly, we distinguish two sampling processes with the stochastic DDPMs and the deterministic DDIMs as $p_s$ and $p_i$. Note the difference between DDPMs and DDIMs only exists in sampling process, while the training remains identical (*i.e.*, a single $p$ may be used in two sampling ways). We use $\mathbf{x}_0'$ to represent the denoised image after editing.

**Sampling vs. Inversion.** We start the analysis of high-dimensional latent spaces of DDMs by considering different sources of latent encodings from the departure space $\epsilon_T$ at the diffusion timestep $T$. Theoretically, the directly sampled latent encodings $\mathbf{x}_T^s$ follow a standard Gaussian distribution $\mathcal{N}(\mathbf{0}, \mathbf{I}_d)$ by definition. In contrast, the inverted latent encodings, as they adopt the DDIMs formulation, **do not** follow an identical trajectory, but the marginal distribution of the forward DDPMs to go from the image space $\mathcal{X}$ to the latent space $\epsilon_T$, leading to a non-standard Gaussian distribution different from the direct sampling. The above describes a **critical statement** that differs from existing work [35], where DDIMs-based inversion has been previously considered as a symmetric process to the generative trajectory.

Table 1: Estimation of the Gaussian radius $r$ for sampled and inverted latent encodings in $\epsilon_T$ from different pre-trained DDMs, datasets and image resolutions (64 and 256).

| Models | Sampling | Inversion | $\triangle r$ |
|---|---|---|---|
| DDPM-CelebA-64 | 110.84 | 95.06 | 15.78 |
| DDPM-LSUN-256 | 443.42 | 430.88 | 12.54 |
| iDDPM-AFHQ-256 | 443.41 | 438.07 | **5.34** |

Table 2: 100-stepwise variation of Gaussian radius for different sample sources and sampling combinations using pre-trained DDM. With different latent encodings and sampling processes, the mixing step appears at approximately the same time around $t = 500$.

| Steps | 1000 | 900 | 800 | 700 | 600 | 500 |
|---|---|---|---|---|---|---|
| $\mathbf{x}_T^s + p_s$ | 0.02 | 0.01 | 0.25 | 0.75 | 1.98 | **4.63** |
| $\mathbf{x}_T^s + p_i$ | 0.02 | 0.02 | 0.21 | 0.74 | 2.03 | **4.76** |
| $\mathbf{x}_T^i + p_s$ | 1.76 | 1.74 | 1.42 | 1.45 | 2.08 | **3.81** |
| $\mathbf{x}_T^i + p_i$ | 1.72 | 1.73 | 1.45 | 1.41 | 2.18 | **3.70** |

We can also empirically demonstrate the above conclusion by estimating the radius $r$ of the high-dimensional Gaussian space. In mathematics, the radius of a high dimensional Gaussian space is defined as the square root of the expected squared distance: $r = \sigma\sqrt{d}$. Specifically, given a $d$-dimensional Gaussian centered at the origin with variance $\sigma^2$. For a point $\mathbf{x} = (x_1, x_2, ..., x_d)$ chosen at random from Gaussian, the expected squared length of $\mathbf{x}$ is:

$$E(x_1^2 + x_2^2 + ... + x_d^2) = dE(x_1^2) = d\sigma^2. \tag{4}$$

For large $d$, the value of the squared length of $\mathbf{x}$ is tightly concentrated about its mean, and the square root of the expected squared distance (namely $\sigma\sqrt{d}$) is called *Gaussian radius*, denoted by $r$.

We calculate the mean of squared length from $N$ samples from $\mathbf{x}_T^s$ and $\mathbf{x}_T^i$ as $\frac{1}{N}\sum_N\sum_j^d((x_{T,j})^2)$. For the sampled latent encoding from $\epsilon_T \sim \mathcal{N}(\mathbf{0}, \mathbf{I}_d)$, the estimated radius $r = \sqrt{d}$, as shown in Tab. 1. In contrast, for the inverted latent encodings from real images, the previously expected radius does not hold. Interestingly, we note that the inverted latent encodings from iDDPMs [41] endure less radius shift compared to DDPMs [21], indicating that this radius estimation may imply the goodness of a pre-trained DDM. The above discussion on the distributions of $\epsilon_T$ helps to better interpret the revealed "distance effect" below. More details about the marginal discussion via DDIMs inversion are included in Appendix A.

**Distance Effect by Deterministic Inversion and Mitigation.** As the next step, we study the influence of an asymmetric inversion trajectory by following the marginal distribution via DDIMs. Previous studies [31, 35] suggest that despite the latent encodings inverted via DDIMs allowing for near-perfect reconstruction, it is rather empirically difficult to directly edit $\mathbf{x}_T^i$ for controlling the final denoised output. Similarly, we observe the phenomena that for the inverted latent encodings $\mathbf{x}_T^i$, a small modification often leads to severely degraded and distorted denoised images following the deterministic sampling $p_i$, as shown in the second row of Fig. 2(a). As a comparison, the directly sampled encoding $\mathbf{x}_T^s$ shows a smooth transition after $p_i$ (see the top row in Fig. 2(a)). This distance effect harms the performance in downstream applications that requires the inversion operation, such as image editing and manipulation tasks where the input is a given raw image $\mathbf{x}_0$ instead of directly sampled latent encoding $\mathbf{x}_T^s$.

Having observed that the inverted latent encoding $\mathbf{x}_T^i$ tends to suffer from distorted denoised images during the interpolation, we aim to further understand and interpret the distance effect from properties of the latent high-dimensional spaces. Intuitively, the sampled latent encodings $\mathbf{x}_T^s$ locate in the area with higher probability concentration mass in $\epsilon_T$ space, while the spatial locations of inverted ones $\mathbf{x}_T^i$, following the marginal diffusion trajectories, have relatively lower concentration mass. This spatial difference remains after the deterministic denoising via $p_i$ due to the homogeneity of intermediate latent spaces [35], resulting in denoised samples from $\mathbf{x}_T^i$ to be rather out-of-distribution and low-fidelity after distance shift in latent spaces. The homogeneity of latent spaces from [35] states that *"the same shift in this space results in the same attribute change in all images"*, while in our context, we consider the spatial difference (*i.e.*, inverted latent encodings have a smaller radius than sampled ones) in terms of Gaussian radius along the denoising trajectory.

From the geometric point of view, the probabilistic concentration mass of a high-dimensional sphere is essentially located within a thin slice at the equator and contained in a narrow annulus at the surface [8]. In addition, one needs to increase the radius of the sphere to nearly $\sqrt{d}$ before there is a

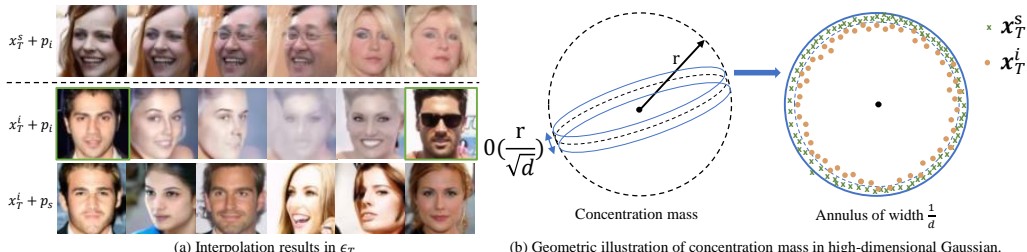

(a) Interpolation results in $\epsilon_T$.  (b) Geometric illustration of concentration mass in high-dimensional Gaussian.

Figure 2: Illustration of distance effect from the qualitative results and geometric properties. (a) The interpolation are conducted on CelabA-64, with different combinations for latent encoding sources and sampling methods. Following the same denoising process $p_i$, inverted latent encodings lead to distorted images. (b) The inversion process reverses images to the latent encodings at the area of less concentration mass with a smaller radius. $\mathcal{O}(\frac{r}{\sqrt{d}})$ is the width of the slice for the concentration mass.

non-zero concentration measure [8]. We explain the cause of the distance effect via the geometric illustration in Fig.2(b). More theoretical details can be found in Appendix B.

In contrast to the deterministic generation that suffers from the distance effect, the stochastic denoising is more robust and helps to alleviate the issue, as shown in Fig. 2(a). Similarly, previous works also reveal that stochasticity is beneficial for the quality of denoised data [28, 31, 35]. Therefore, adding stochasticity is a mitigation solution for the distance effect. From the perspective of spatial properties, the stochasticity brings the inverted latent encodings $\mathbf{x}_T^i$ from the area with lower concentration mass to higher ones, thus making the denoising process more robust.

## 4 Mixing Step in Diffusion Models

**Problem Formulation.** In the mathematical and statistical study of the Markov process, the mixing time is defined as a key step $t$, at which the intermediate distribution converges to the stationary distribution [38] [2]. Inspired by the established mathematical formulations, we introduce the mixing step problem for diffusion models as the search for the critical diffusion step $t \in \{1, ..., T\}$, where the Gaussian distribution converges to the final data distribution in the reverse denoising process under some pre-defined distance measures (usually in *total variation distance*, noted as $|| \cdot ||_{TV}$). It is worth noting that, unlike the distance effect which is induced by the deterministic inversion method, the mixing step is a characteristic carried by **generic DDMs**, as we show that the mixing step appears at the same time both for the directly sampled latent encodings and the inverted ones. In practice, we show in the experiments in Sec. 6 that imposing editing and manipulations on the mixing step is more effective for downstream tasks. Meanwhile, similar concepts have been empirically studied as a hyper-parameter as "return step" [31] or "editing step" [35] without further investigations into its theoretical meanings. Proof and details of the following property can be found in the supp.

*Property* 4.1. Under the total variation distance measure $|| \cdot ||_{TV}$, the mixing step $t_m$ for a DDM with data dimensionality $d$ is formed during training (i.e., irrelevant to the sampling methods). $t_m$ is mainly related to the transition kernels, the stationary distribution (i.e., datasets), and the dimensionality $d$. In practice, a radius shift of approximately 4 can be considered as a searching criterion for the mixing step in the denoising chain.

**Automatic Search via Radius Estimation.** While the rigorous theoretical deviation for the mixing step $t_m$ depends on many factors and is a complex and non-trivial problem (see our supp. for details), we present here an effective yet simple method to search for the mixing step by estimating the radius $r$ of latent high-dimensional Gaussian spaces. We can estimate the radius of the latent spaces along the denoising trajectory and show that the change of $r$ (corresponding to the transition kernel in Property 4.1) reveals the appearance of the mixing step for a pre-trained DDM. Our proposed method automates the search for this critical diffusion step, while it has been previously determined by manually checking the quality of final denoised images [31, 35]. More detailed discussion and proof about the empirical search method can be found in Appendix D.

---

[2]See more details about the Markov mixing time in Appendix C.

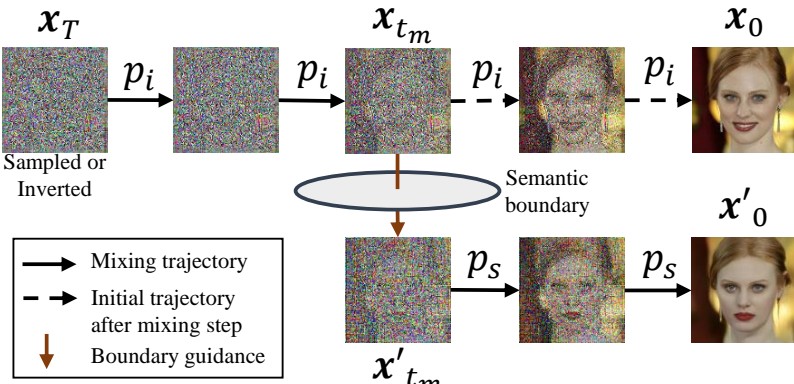

Figure 3: ***BoundaryDiffusion* for semantic control in one-step editing.** We propose to guide the initial deterministic generative trajectory via the semantic boundary at the mixing step $t_m$ for image manipulation.

Specifically, we follow the procedure described in Sec. 3, where we calculate the mean of the squared distance of a latent sample from the origin as the estimation of the Gaussian square. We compute the radius of the latent encodings following the generative trajectory, the earliest step with the largest radius shift ($\triangle r \approx 4$) is an approximation of the mixing step. We show the change of radius estimation for different combinations of latent encoding sources and sampling methods, demonstrating the mixing step is a generic characteristic irrelevant to the above factors in Tab. 2.

## 5 *BoundaryDiffusion* for Semantic Control

In this section, we propose our *BoundaryDiffusion* method to achieve **one-step** semantic control and editing by guiding the denoising trajectory using the semantic boundaries in high dimensional latent spaces with pre-trained and frozen DDMs. Specially, we leverage the stochasticity and the mixing step $t_m$ for better qualitative results and more effective manipulation.

**Semantic Boundary Search.** In the case of attribute manipulation with given annotations (*e.g.*, face attributes modifications), we propose to search for the semantic separation hyperplane as a classification problem as in [51]. Specifically, we use the linear SVM [18] to classify the latent encodings using the label annotations of their corresponding raw data, and the parameters in SVM are used as the hyperplanes. The validation of using raw data ground truth annotation at the latent space level is guaranteed by the deterministic inversion and reconstruction via DDIMs [54]. We consider this application scenario to be conditional semantic editing, which allows us to edit a semantic attribute given a real raw image. It is worth noting that the objective of using SVM is to fit the *existing semantic hyperplanes* rather than learning classifiers or networks.

In addition to the above application scenario, we can further extend our semantic boundary search to more flexible settings such as text-based image editing by leveraging the popular large-scale cross-modality generative models such as DALLE-2 [46] and Imagen [20]. Particularly, we can easily generate high-fidelity images based on a given textual prompt and utilize the synthetic images as samples to search for the semantic boundary using the same classification method.

An extra application for our semantic boundary searching is to achieve semantic control for unconditional image synthesis, due to the reason that the semantic boundary is a feature learned and formed during the training. In contrast, other learning-based methods like DiffusionCLIP [31] that change the original parameters of pre-trained DMMs are not suitable for this downstream task.

**Mixing Trajectory for Semantic Control.** After having located the semantic boundaries, we propose to achieve semantic manipulation by guiding the trajectory via the identified boundaries using the projected distance control. We note the unit normal vector of the hyperplane as $\mathbf{n} \in \mathbb{R}^d$, and define the sign-sensitive "distance" of the latent encoding $\mathbf{x}_{t_m}$ to the boundary similar to [51]:

$$d(\mathbf{n}, \mathbf{x}_{t_m}) = \mathbf{n}^T \mathbf{x}_{t_m}. \tag{5}$$

Meanwhile, we add the stochasticity after the editing the latent encodings at the mixing step $t_m$. Overall, the mixing trajectory with boundary guidance can be summarized as:

$$p_{mix}(\mathbf{x}_{t-1}|\mathbf{x}_t) = \begin{cases} p_i(\mathbf{x}_{t-1}|\mathbf{x}_t) & \text{if } T \geq t > t_m \\ p_s(\mathbf{x}'_{t-1}|\mathbf{x}'_t) & \text{if } t_m \geq t \end{cases} \tag{6}$$

where $\mathbf{x}'_{t_m} = \mathbf{x}_{t_m} + \zeta d(\mathbf{n}, \mathbf{x}_{t_m})$, and $\zeta$ is the editing strength parameter that controls the projected distance to the target semantic boundary. We illustrate our proposed boundary-guided mixing trajectory method in Fig. 3. Note this operation can also be extended to multi-attribute semantic control, either by iteratively applying guidance to multiple boundaries (which is a linear operation), or by modifying Eq. 5 to $d(\mathbf{N}, \mathbf{x}_{t_m}) = \mathbf{N}^T \mathbf{x}_{t_m}$, where $\mathbf{N} = [\mathbf{n_1}, \mathbf{n_2}, ..., \mathbf{n_m}]$ for $m$ different attributes. The concrete algorithm of our proposed *BoundaryDiffusion* is presented in Appendix F.

## 6 Experiments

### 6.1 Experimental Setup

**Task Settings.** We conduct our experiments on multiple variants of semantic control tasks, including *real image conditioned semantic editing*, *real image conditioned text-based editing*, and *unconditional image synthesis with semantic control*. Among those tasks, the first two scenarios are conditioned on given images and therefore involve the inversion process, while the last one is an extension application for image synthesis without given image input.

**Model Zoo and Datasets.** We test different pre-trained DDMs on various datasets with different image resolutions for experiments. Specifically, we test the DDPMs on the CelabA-64 [39], CelebA-HQ-256 [27], LSUN-Church-256 and LSUN-Bedroom-256 [66]. We also experiment with pre-trained improved DDPM [41] on the AFHQ-Dog-256 [10]. In particular, we emphasize here that the experiments with different image resolutions are critical in our work, especially for the high-dimensional space analysis since the resolutions represent the dimensionality of the latent spaces as shown in Tab. 1 and Fig. 2. However, in all the editing experiments, we use a resolution of $256 \times 256$ as the default setting for better visualization quality.

**Operational Latent Spaces.** Previous works have either operated on the $\epsilon$-space [31] or on the $h$-space from the bottleneck layer of the U-Net [35], showing that $h$-space has better semantic meanings with homogeneity, linearity, and robustness [35]. In our main experiments, we impose the guidance on both latent encodings in both $\epsilon$ and $h$ levels. All the boundary guidance is imposed in a **single-step** at the mixing step $t_m$. We show in our experimental results later that the single-step operation is more effective at the $\epsilon$-level, while the $h$-level may require iterative guidance at multiple diffusion steps.

**Implementations.** We use the linear SVM classifier for searching the semantic boundary. We implement the SVM via the sklearn python package with the number of parameters equal to the total dimensionality of the latent spaces. For $\epsilon$-space, the dimensionality $d_\epsilon = 3 \times 256 \times 256 = 196,608$. For the $h$-space, the dimensionality depends on the pre-trained DDMs architecture implementation for the U-Net [49]. In our experiments, we use the same level of latent spaces as in [35], which have a dimensionality of $d_h = 8 \times 8 \times 512 = 32,768$. In practice, we observe approximately 100 images are sufficient for finding an effective semantic boundary. For the text-based semantic editing scenario, we use synthetic images generated from Stable Diffusion using text-prompt [48]. These image samples can be obtained via any other pre-trained text-to-image generative models.

In practice, the hyperplanes are found via linear SVMs [18], with almost negligible learning time of about 1 second on a single RTX3090 GPU. For the inference, the time cost remains at the same level as other SOTA methods. Specifically, by using the skipping step techniques, we can already generate high-quality denoised images using approximately 40-100 steps, which take from 1.682 - 13.272 seconds, respectively on a single RTX-3090 GPU.

**Comparison and Evaluations.** We mainly compare our proposed method with the most recent state-of-the-art image manipulation works with diffusion models: the DiffusionCLIP [31] and the Asyrp [35]. Both works consist of extra learning processes. Our evaluations include quantitative and qualitative assessments for different experimental settings, similar to most existing generation works. For the inversion and reconstruction part, since we adopt the same technique as in [31, 35] via DDIMs [54] and achieve near-perfection reconstruction results, we show qualitative samples in Fig. 1. As for the conditional editing, we show quantitative results on the CelebA-HQ using

Table 3: Evaluation results on CelebA-HQ-256 for real image conditioned semantic editing. FID scores are reported on the test set with 500 raw images, averaged on *"add or remove smile"* editing. The user study follows similar evaluation questions in [35].

| Methods | $S_{dir}$ ↑ | SC ↑ | ID ↑ | FID ↓ | User Quality ↑ | User Attribute ↑ |
|---------|-------------|------|------|-------|----------------|------------------|
| StyleCLIP [43] | 0.13 | 86.8% | 0.35 | - | - | - |
| StyleGAN-NADA [15] | 0.16 | 89.4% | 0.42 | - | - | - |
| DiffusionCLIP [31] | 0.17 | **93.7%** | 0.70 | 86.23 | 3.2% | 8.2% |
| Asyrp [35] | **0.19** | 87.9% | - | 68.38 | 41.3% | 44.9% |
| Ours BoundaryDiffusion | 0.17 | 90.4% | **0.73** | **63.14** | **55.5%** | **46.9%** |

three metrics: Directional CLIP similarity ($S_{dir}$) computed via CLIP [45], segmentation consistency (SC) computed via [65, 68, 67], and face identity similarity (ID) using [13]. We also report the FID scores [19] of edited images on the testing set of CelebA-HQ [27]. We further conduct human studies as subjective evaluations. Additionally, we also report the classification accuracy for the semantic boundary separation validation as in Appendix G.

## 6.2 Experimental Results

**Conditional Semantic Manipulation.** We show the qualitative results for conditional semantic manipulation on the CelebA-HQ-256 [27] using the pre-trained DDPM [21] in the top row of Fig. 1. The quantitative evaluations are shown in Tab. 3, where we achieve SOTA performance. In particular, we emphasize the fact both SOTA methods [31, 35] directly use the directional CLIP loss as part of the loss function to align the direction between the embeddings of the reference and generated image in the CLIP space [45] in their learning processes, which helps to increase the score of $S_{dir}$. Additionally, they also optimize the identity loss $\mathcal{L}_{id}(x_0', x_0)$ to preserve the facial identity consistency between the original image $x_0$ and the edited one $x_0'$, thus boosting the ID score. While our proposed *BoundaryDiffusion* method does not use any learning loss function for semantic control, yet achieves very competitive scores in all the metrics.

**Text-Guided Semantic Manipulation.** With the success of large-scale cross-modality generative models [46, 20], we are able to extend the proposed *BoundaryDiffusion* method to other downstream tasks such as text-guided semantic manipulations. Qualitative examples are presented in the middle part of Fig. 1, where we show the original image input, reconstruction results, and the edited samples. We also perform experiments on the unseen domain transfer and show the example of editing a *dog* to a *fox*, as indicated using the red text prompt in Fig. 1. We note one limitation of our *BoundaryDiffusion* is relatively difficult for unseen domain editing tasks compared to other learning-based methods, which aligns with the expectation since the frozen DDMs do not have the knowledge for unknown domain distributions.

**Unconditional Semantic Control.** One unique application scenario we present in the bottom row of Fig. 1 is to apply the identified semantic boundary to the unconditional synthesis setting, where the departure latent encoding $\mathbf{x}_T$ is now directly sampled from a standard Gaussian without a given real image as input. Since the semantic boundary is an intrinsic characteristic formed during the training process of DDMs, we can easily re-use it for semantic control under a general unconditional denoising trajectory. In comparison, the existing SOTA methods that modify the parameters of pre-trained DDMs during additional learning are no longer applicable to this task.

**Ablation on Operational Latent Spaces and Editing Strength.** We can easily control the editing strength via the editing distance as defined in Sec. 5. Fig. 4 shows an example for editing the *smile* attribute on CelebA-HQ. We use the hyper-parameter $\zeta$ to control the degree of modification. From the same figure, we observe that we achieve a stronger editing effect at the same distance when applying the boundary guidance to both latent levels at $\epsilon_{t_m}$ and $h_{t_m}$.

**User Study.** We also conducted a subjective user study to compare the performance with Diffusion-CLIP [31] and Asyrp [35], where we interviewed 20 human evaluators to rank the edited images in terms of general quality and attribute editing effects. Specifically, we asked the evaluators to pick the best-edited result in terms of two aspects: **1).** General quality: which image quality do you think is the best (clear, fidelity, photorealistic)? **2).** Attribute: which image do you think achieves the best attribute editing effect (natural, identity preservation with respect to the given raw image)? Tab 3

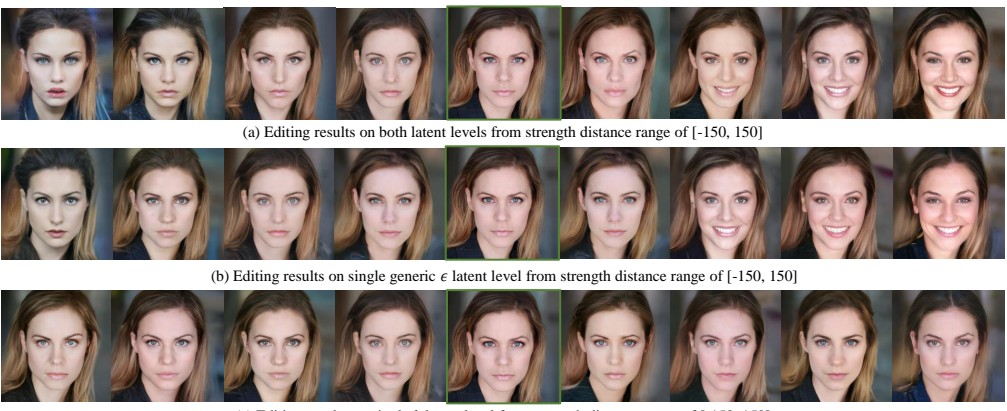

(a) Editing results on both latent levels from strength distance range of [-150, 150]

(b) Editing results on single generic $\epsilon$ latent level from strength distance range of [-150, 150]

(c) Editing results on single $h$ latent level from strength distance range of [-150, 150]

Figure 4: **Ablation on editing strength control via distance with different operational spaces.** We can modify the editing strength by changing the distance between the latent encoding $\mathbf{x}_{t_m}$ to the target boundary. We show that the one-step guidance is more effective when applied in the $\epsilon$-level space. Images in green boxes denote the given raw image.

shows the percentage of each method picked as the best result. Details about the user study and more **randomly selected, non-cherry-picked** qualitative samples are included in Appendix G.

## 7 Discussion and Conclusion

Our work explores and leverage the great potential of pre-trained unconditional diffusion models without learning any additional network model modules. We provide a rather comprehensive analysis from the high-dimensional space perspective to better understand the latent spaces and the trajectory for pre-trained DDMs. We introduce the mixing step for DDMs, which characterizes the convergence of generic DDMs, and present a simple yet effective method to search for the mixing step via Gaussian radius estimation in high-dimensional latent spaces. Last but not least, we propose *BoundaryDiffusion*, an effective learning-free method with boundary-guided mixing trajectory to realize effective semantic control and manipulation for downstream tasks using multiple pre-trained DDMs and datasets.

**Broader Impact and Limitation.** We discuss the broader impact of this work. Firstly, the primary goal of this work is not to create new generative models or generate synthetic data, but to explore the potential of the current generative models for better usage. To do so, we also propose a new perspective to better understand and interpret the DDMs, which is the analysis of high-dimensional latent space behaviors using the theoretical tools from mathematics and statistics. In the meanwhile, during the process of exploring and separating the semantic boundary, we leverage the current popular cross-modality generative models to synthesize images with a text prompt. However, all the generated images are only used for boundary detection. We believe our work brings valuable insights to the research community in terms of a better understanding and further exploration via training-free methods to apply diffusion generative models.

As for limitations, since our *BoundaryDiffusion* method is learning-free and does not introduce any extra parameters to the base diffusion models, the ability for unseen domain image editing is relatively limited compared to other learning-based methods.

## Acknowledgements

This work is supported by NSF IIS-2309073. This article solely reflects the opinions and conclusions of its authors and not the funding agency. The first author of this paper would also like to thank Dr. Alain CHILLÈS for the insightful discussions on the theoretical and mathematical analysis.

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

We present the detailed related work about the denoising diffusion models including the marginal discussion in Appendix A. The high-dimensional space properties and lemmas are introduced in Appendix B, and we explicitly describe how those established theoretical theorems are used and connected to our analysis in the main paper Sec. 3. In Appendix C, we provide the theoretical foundations of the Markov mixing study, which inspires us to formulate the mixing step problem for DDMs. More details about the mixing step problem, formulation, proof and discussion are included in Appendix D. Appendix E includes details about our semantic boundary search method and further discussions in terms of two latent space levels (*e.g.*, generic $\epsilon$-space and $h$-space from the U-Neu bottleneck [35]). We show the algorithm of our proposed boundary-guided mixing trajectory method in Appendix F. More **randomly selected and non cherry-picked** experimental results, details about user study, and some failure cases analysis are shown in Appendix G.

## A    Detailed Related Work

### A.1    Denoising Diffusion Models

While we have briefly introduced the preliminaries on DDPMs [21] and DDIMs [54] in the main paper, we re-organize and present more details here. We note that the relevant background is mainly from the original papers, we only include the relevant background information to better illustrate our ideas in this work.

The key idea for generative tasks is to approximate a data distribution $q(x_0)$ with a model learned distribution $p_\theta(x_0)$ that can be easily sampled from. The original Denoising Diffusion Probabilistic Models (DDPMs) [53] propose to use latent variable models to fulfill the goal with the following specific form:

$$p_\theta := \int p_\theta(x_{0:T})dx_{1:T}, \tag{7}$$

where $x_1, ..., x_T$ are variables modeled by the latent states of a Markov chain, which have the same dimensionality as the actual data $x_0 \sim q(x_0)$. Specifically, we have:

$$p_\theta(x_{0:T}) := p_\theta(x_T) \prod_{t=1}^{T} p_\theta^{(t)}(x_{t-1}|x_t). \tag{8}$$

The training objective is the variational lower bound on negative log likelihood:

$$
\begin{aligned}
L &:= \mathbb{E}[-\log p_\theta(x_0)] \\
&\leq \mathbb{E}[-\log \frac{p_\theta(x_{0:T})}{q(x_{1:T}|x_0)}] \\
&= \mathbb{E}_q[-\log p(x_T) - \sum_{t \geq 1} \log \frac{p_{\theta(x_{t-1}|x_t)}}{q(x_t|x_{t-1})}].
\end{aligned}
\tag{9}
$$

The above formulation indicates that the DDPMs can be learned with a pre-defined inference procedure $q(x_{1:T}|x_0)$. In the case of [21], the authors propose to model the Markov chain with Gaussian transitions parameterized by a decreasing sequence $\alpha_{1:T} \in (0, 1]^T$ as follows:

$$q(x_{1:T}|x_0) := \prod_{t=1}^{T} q(x_t|x_{t-1}), \tag{10}$$

where $q(x_t|x_{t-1}) := \mathcal{N}(\sqrt{\frac{\alpha_t}{\alpha_{t-1}}}x_{t-1}, (1 - \frac{\alpha_t}{\alpha_{t-1}})\mathbf{I})$.

We often refer to the above-mentioned processes from $x_0$ to $x_T$ and from $x_T$ to $x_0$ as *forward process* and *reverse process* (or *generative process*), respectively. Intuitively, the forward process adds noise to data $x_0$, while the reverse process denoises a noisy latent variable $x_{1:T}$. The reverse denoising is stochastic based on this formulation.

### A.2    Marginal Discussion for Deterministic Inversion

Motivated to reduce the iteration numbers from the original DDPMs [53, 21], Denoising Diffusion Implicit Models (DDIMs) [54] propose to generalize the inference process (*i.e.*, forward process) from

a Markov chain to a Non-Markov one. The theoretical support for the proposed generalization lies within the fact the learning objective of DDPMs only depends on the conditional (on $x_0$) marginals $q(x_t|x_0)$, instead of the conditional (on $x_0$) joint $q(x_{1:T}|x_0)$.

Based on the previous fact, DDIMs consider a family of inference distribution $\mathcal{Q}$, indexed by a real vector $\sigma \in \mathbb{R}^T_{\geq 0}$:

$$q_\sigma(x_{1:T}|x_0) := q_\sigma(x_T|x_0) \prod_{t=2}^{T} q_\sigma(x_{t-1}|x_t, x_0), \tag{11}$$

where $q_\sigma(x_T|x_0) = \mathcal{N}(\sqrt{\alpha_T}x_0, (1-\alpha_T)\mathbf{I})$. Specifically, the $q_\sigma(x_{t-1}|x_t, x_0)$ is carefully designed in a way that the mean function satisfies the above Gaussian kernel as:

$$q_\sigma(x_{t-1}|x_t, x_0) = \mathcal{N}(\sqrt{\alpha_{t-1}}x_0 + \sqrt{1 - \alpha_{t-1} - \sigma_t^2}\frac{x_t - \sqrt{\alpha_t}x_0}{\sqrt{1-\alpha_t}}, \sigma_t^2\mathbf{I}). \tag{12}$$

Using the Bayes' rule, Eq. 12 can be further rewritten as:

$$q_\sigma(x_t|x_{t-1}, x_0) = \frac{q_\sigma(x_{t-1}|x_t, x_0)q_\sigma(x_t|x_0)}{q_\sigma(x_{t-1}|x_0)}. \tag{13}$$

The above Eq. 12 and Eq. 13 show that the Non-Markov process $q_\sigma$ considered in DDIMs is marginal and also Gaussian (but not a standard one).

After having specified the forward process, DDIMs propose a different variant of the sampling process where the model is expected to first predict the corresponding noiseless $x_0$ given a noisy observation $x_t$, and use the prediction to obtain $x_{t-1}$ through Eq. 13. Specifically, the iteration can be written as follows:

$$x_{t-1} = \sqrt{\alpha-1}(\frac{x_t - \sqrt{1-\alpha_t}\varepsilon_\theta^{(t)}(x_t)}{\sqrt{\alpha_t}}) + \sqrt{1 - \alpha_{t-1} - \sigma_t^2}\varepsilon_\theta^{(t)}(x_t) + \sigma_t\varepsilon_t, \tag{14}$$

where $\varepsilon_t \sim \mathcal{N}(\mathbf{0}, \mathbf{I})$. By choosing the $\sigma_t = 0$ for all steps $t$, the random noise induced by the last term from Eq. 14 is removed, and therefore changing the stochastic process from the original DDPMs formulation to a deterministic one.

By connecting the Eq. 14 to the Euler integration for solving ordinary differential equations (ODEs), it can be further rewritten as:

$$\frac{x_{t-1}}{\sqrt{\alpha_{t-1}}} = \frac{x_t}{\sqrt{\alpha_t}} + \left(\sqrt{\frac{1-\alpha_{t-1}}{\alpha_{t-1}}} - \sqrt{\frac{1-\alpha_t}{\alpha_t}}\right)\epsilon^t(x_t), \tag{15}$$

which is the Euler solution for the following:

$$d\bar{x}(t) = \varepsilon_\theta^{(t)}(\frac{\bar{x}(t)}{\sqrt{\sigma^2+1}})d\sigma(t). \tag{16}$$

Therefore, when we adopt the deterministic inversion method to convert $x_0$ to $x_T$, we preserve the marginal property of the considered family of inference distribution $\mathcal{Q}$. Intuitively, extending this distribution family to the $d$-dimensional space, it ensembles a group of Gaussian distributions parameterized by the defined $\alpha$ sequence. Given the denoising process can be considered as a trajectory, the deterministic inversion follows and stays at the border of the space ensemble.

### A.3   Other Related Works

Other studies related to this work include the areas of GANs inversion, image editing and manipulations.

**GAN Inversion.** GAN inversion problem [62] is proposed to tackle the lack of inference capability in GANs [16]. As another powerful model other than DMs for data generation, many GAN models [27, 9, 29, 30] have been proposed for high-quality image synthesis. With high-level objectives to invert a given image and to apply it in downstream tasks like image editing, the two problems are often studied separately. Specifically, due to the intractability of GAN generation, many works have been

focused solely on the first objective to invert an image to the latent space and to reconstruct from the latent encoding, which corresponds to the initial and primary goal of GAN inversion. There are three main technical directions for the inversion and reconstruction problem, which consists of learning an additional deterministic encoder [70, 47, 61, 3], directly solving the optimization problem [1, 2, 25, 12], or a hybrid way that combines the above two techniques [69, 7, 6].

Different from existing GAN inversion works, we leverage the better tractability of DMs and use the deterministic property from the denoising diffusion implicit models (DDIMs) [54] to achieve the inversion and reconstruction when studying the diffusion direction.

**Image Manipulation and CLIP Guidance.** Image manipulation based on generative models mainly covers two categories. While one branch of existing works often requires retraining of a generative model (*e.g.*, GANs [16]) [42, 59, 36, 5], others are studied as a downstream task application for GAN inversion works [69, 51, 29, 1]. For image manipulation using the GAN inversion technique, a prerequisite for effective editing is a disentangled understanding of latent spaces from pre-trained GAN models. The analysis on the latent space addresses several different separate latent spaces such as the $\mathcal{Z}$ space for generic GANs [16] and the $\mathcal{W}$ space from StyleGAN [29]. The current SOTA methods for diffusion-based editing like DiffusionCLIP [31] and Aysrp [35] all adopt the CLIP guidance as part of their loss function during the learning process.

In this work, we adopt a similar semantic disentanglement idea as the tool to interpret and understand the latent space along the chain. At the same time, we are able to leverage our analysis and a better understanding of the latent space to achieve real-face image editing.

## B  High Dimensional Space

In this section, we provide the necessary theoretical foundations for understanding the geometric and probabilistic properties of high-dimensional spaces. The majority of the properties and lemmas we describe here are established theorems from high-dimensional space studies in mathematics and statistics from [8]. We omit the detailed proofs for the following properties and lemmas, and kindly ask readers to refer to the original book if interested.

*Property* B.1. For a unit-radius sphere in high dimensions, as the dimension $d$ increases, the volume of the sphere goes to 0, and the maximum possible distance between two points stays at 2.

**Lemma B.2.** *The surface area $A(d)$ and the volume $V(d)$ of a unit-radius sphere in d-dimensions are given by:*

$$A(d) = \frac{2\pi^{d/2}}{\Gamma(d/2)}, V(d) = \frac{\pi^{d/2}}{\frac{d}{2}\Gamma(d/2)}, \tag{17}$$

where $\Gamma(x)$ is a generalization of the factorial function for noninteger values of $x$.

The above Property B.1 and Lemma B.2 are generic geometric properties for high-dimensional spheres, but also applicable to high-dimensional Gaussian in which we are interested in the context of DDMs. To draw the connections with our context for studying the latent spaces of DMMs, with higher dimensionality, the latent Gaussian spaces of pre-trained DDMs become more difficult to operate due to decreased volume and mass concentration, as empirically suggested in [31, 35].

*Property* B.3. The volume of a high-dimensional sphere is essentially all contained in a thin slice at the equator and is simultaneously contained in a narrow annulus at the surface, with essentially no interior volume. Similarly, the surface area is essentially all at the equator.

The Property B.3 implies the connection with the standard Gaussian in $\epsilon_T$ from direct sampling. In Fig.2 (b) of the main paper where we illustrate the geometric and probabilistic properties of samples in the $\epsilon_T$ space, the inverted ones locate in the inner border area of the narrow annulus, which is also empirically verified in our Tab. 1 in the main paper. As those inverted latent encodings have a smaller radius than the expected standard Gaussian case.

**Lemma B.4.** *For any $c > 0$, the fraction of the volume of the hemisphere above the plane $x_1 = \frac{c}{\sqrt{d-1}}$ is less than $\frac{2}{c}e^{-\frac{c^2}{2}}$.*

The above Lemma B.4 explains the volume range we show in Fig.2 (b) of the main paper in the left side of the Gaussian sphere to show the concentration mass, which is in the order of $O(\frac{r}{\sqrt{d}})$.

**Lemma B.5.** *The maximum likelihood spherical Gaussian for a set of samples is the one over center equal to the sample mean and standard deviation equal to the standard deviation of the sample.*

The above Lemma B.5 provides the theoretical justifications for using the mean of squared distance to estimate the radius of Gaussian high-dimensional space.

## C   Markov Mixing

The mixing time defines a parameter that measures the time required by a Markov chain for the distance to stationary to be small [38]. The study of Markov mixing time aims to quantify the speed of convergence for Markov chains, and requires some other necessary preliminary knowledge on the *total variance distance* and the Convergence Theorem, which we will briefly describe below.

Firstly, to quantify the convergence characteristic of Markov chains, an appropriate distance measure metric is a prerequisite. In the literature, the *total variation distance* is the metric used to define the distance.

**Definition C.1.** The total variation distance between two probability distributions $\mu$ and $\nu$ on $\mathcal{X}$ is defined by:

$$||\mu - \nu||_{TV} = \max_{A \subseteq \mathcal{X}} |\mu(A) - \nu(A)|. \tag{18}$$

In the above definition, $A$ is a probabilistic event, indicating that the distance between $\mu$ and $\nu$ is the maximum difference between probabilities assigned to a single event by the two distributions. This initial definition is not very practical to estimate the actual distance, which further induces the following propositions.

**Proposition C.2.** *Let $\mu$ and $\nu$ to be two probability distributions on $\mathcal{X}$. Then*

$$||\mu - \nu||_{TV} = \frac{1}{2} \sum_{x \in \mathcal{X}} |\mu(x) - \nu(x)|. \tag{19}$$

*Proof.* Let $B = \{x : \mu(x) \geq \nu(x)\}$ and let $A \subset \mathcal{X}$ be any event. Then we have

$$\mu(A) - \nu(A) \leq \mu(A \cap B) \leq \mu(B) - \nu(B). \tag{20}$$

The first inequality holds since any $x \in A \cap B^c$ satisfies $\mu(x) - \nu(x) < 0$, and thus the difference in probability cannot decrease when such elements of $B$ are eliminated. For the second inequality, we note that including more elements of $B$ can not decrease the difference in probability.

By the same reasoning, we have:

$$\nu(A) - \mu(A) \leq \nu(B^c) - \mu(B^c). \tag{21}$$

The upper bounds on the right sides of Equation (20) and (21) are the same. Furthermore, by taking $A = B$ or $A = B^c$, then $|\mu(A) - \nu(A)|$ is equal to the upper bound. Therefore, we arrive at:

$$||\mu - \nu||_{TV} = \frac{1}{2}(\mu(B) - \nu(B) + \nu(B^c) - \mu(B^c)) = \frac{1}{2} \sum_{x \in \mathcal{X}} |\mu(x) - \nu(x)|. \tag{22}$$

$\square$

The above proposition reduces total variance distance to a simple sum over the state space, which is an important theoretical support to formulate our mixing step problem and empirical search method. The proof process also reveals the following remark.

*Remark* C.3. $||\mu - \nu||_{TV} = \sum_{x \in \mathcal{X}, \mu(x) \geq \nu(x)} [\mu(x) - \nu(x)]$.

We then proceed to introduce the convergence theorem, which claims that aperiodic Markov chains converge to their stationary distributions at a key step, which is the direct theoretical foundation for us to introduce the mixing step problem for DDMs.

*Theorem* C.4. **Convergence Theorem** Suppose that $P$ is irreducible and aperiodic, with stationary distribution $\pi$. Then there exit constants $\alpha \in (0, 1)$ and $C > 0$ such that:

$$\max_{x \in \mathcal{X}} ||P^t(x, .) - \pi||_{TV} \leq C\alpha^t. \tag{23}$$

There exist multiple mathematical versions for the proof of the convergence theorem, which we omit in this appendix. Note that the assumptions for $P$ to be irreducible and aperiodic are essential. We recall here the definition of an *irreducible* chain $P$.

**Definition C.5.** A chain $P$ is called irreducible if for any two states $x, y \in \mathcal{X}$, there exists an integer $t$ (possibly depending on $x$ and $y$) such that $P^t(x, y) > 0$.

Intuitively, this means that it is possible to get from any state to any other state using only transitions of positive probability. This is verified in the current formulation of DDMs, indicating that DDMs satisfy the pre-requite to be an irreducible Markov chain.

Next, we recall the definition of the period for a Markov chain.

**Definition C.6.** Let $\tau(x) := \{t \geq 1 : P^t(x, x) > 0\}$ be the set of times when it is possible for the chain to return to starting position $x$. The period of state $x$ is defined to be the greatest common divisor of $\tau(x)$.

For an irreducible chain, the period of the chain is defined to be the period that is common to all states, and the chain is aperiodic if all states have period 1. Intuitively, the above definition and property match the actual formulation and implementations of DDMs, given the fact that plenty of existing DDMs [54, 4, 17, 71] propose an auxiliary loss to predict directly the denoised $x_0$ at arbitrary step.

Having introduced the above definitions, we are now ready to present the formal definition of mixing time in Markov chain studies.

**Definition C.7. Definition of Mixing Time** The mixing time is a parameter that measures the time required by a Markov chain for the distance to the stationary distribution to be small, following the definition below:

$$t_{mix}(\epsilon) := \min\{t : d(t) \leq \epsilon\} \text{ and } t_{mix} := t_{mix}(1/4). \tag{24}$$

In particular, taking $\epsilon = \frac{1}{4}$ above yields

$$d(lt_{mix}) \leq 2^{-l} \text{ and } t_{mix}(\epsilon) \leq \lceil log_2 \epsilon^{-1} \rceil t_{mix}. \tag{25}$$

In addition to the initial definition of mixing time, we also need the background knowledge on the *time reversal* to search for the actual mixing step in a more practical way.

**Definition C.8.** For a distribution $\mu$ on a group $G$, the reversed distribution $\hat{\mu}$ is defined by $\hat{\mu}(g) := \mu(g^{-1})$ for all $g \in G$.

The time reversal is directly related to the two-direction design of DDMs, and ensures that the mixing step remains at **a fixed position** in two directions for both diffusion and generative processes. This property is also critical to better understand the DDMs, and provides theoretical justifications for us to search for the mixing step along the generative direction using the Gaussian radius estimation. In fact, the Gaussian radius estimation search method can only be valid and applied in the generative direction but not the inverse diffusion process. The reasons are discussed in the following section when we show more empirical results.

**Lemma C.9.** *Let $P$ be the transition matrix of a random walk on a group $G$ with increment distribution $\mu$ and let $\hat{P}$ be that of the walk on $G$ with increment distribution $\hat{\mu}$. Let $\pi$ be the uniform distribution on $G$. Then for any $t \geq 0$,*

$$||P^t(id, \cdot) - \pi||_{TV} = ||\hat{P}^t(id, \cdot) - \pi||_{TV}. \tag{26}$$

The lemma above implies the remark below, which will be used in our proof for the Property 4.1.

*Remark* C.10. If $t_{mix}$ is the mixing time of a random walk on a group and $t_{\hat{mix}}$ is the mixing time of the reversed walk, then $t_{mix} = t_{\hat{mix}}$.

We hereby finish introducing the necessary background on Markov mixing studies, and continue to a more detailed discussion of the mixing step problem of DDMs.

# D  More Discussion on Mixing Step

Inspired by the Markov mixing studies, we remark that the current formulation of DDMs satisfies several key assumptions as described in Appendix C, including most importantly, DDMs model an irreducible and aperiodic chain. Note that our current exploration and formulation for the mixing step of DDMs are not absolutely thorough and complete, which can be considered as an approximate and adapted version of the mathematical Markov mixing time.

## D.1  Proof for Property of Mixing Step

We rewrite the Property of mixing step for DDMs here before going to the detailed proof.

*Property* D.1.  Under the total variation distance measure $||\cdot||_{TV}$, the mixing step $t_m$ for a DDM with data dimensionality $d$ is formed during training (i.e., irrelevant to the sampling methods). $t_m$ is mainly related to the transition kernels and the stationary distribution (i.e., datasets), and less dependant on the dimensionality $d$.

*Proof.*  The proof for the above property consists of several steps.

*Existence justification.*  Firstly, we have shown that DDMs model a group of chains that are irreducible and aperiodic, and thus the convergence theorem holds for DDMs. This fact establishes the theoretical foundation to find such a critical convergent step that theoretically characterizes the convergence of pre-trained DDMs.

*Directions to approach.*  Secondly, we show that the mixing step is large and mostly dependent on the transition kernel. Here, we have to clarify the direction of the DDMs we are tackling. Fortunately, based on the time reversal from Lemma C.9 and Remark C.10, whichever direction gives the same mixing step, provides us with the flexibility to study either direction. However, in practice, the easiest way to approach the mixing step is to theoretically infer the transition kernel in the diffusion direction, and then empirically search for it in the denoising direction. We will first provide the method and explain the reasons for such a design.

*Theoretical based transition kernel study.*  We hereby restrict ourselves in considering the diffusion process. Given pre-trained DDMs, according to Lemma C.9, we have an irreducible transition matrix $P$ on space $\mathcal{X}$. In the current scenario of diffusion direction from $\mathbf{x}_0$ to $\mathbf{x}_T$, the stationary distribution is the standard Gaussian $\mathcal{N}(0, \mathbf{I}_d)$ in $\epsilon_T$. The transition matrix is a pre-defined Gaussian with known mean value and variance, thus we have

$$P = q(\mathbf{x}_t|\mathbf{x}_{t-1}) := \mathcal{N}(\mathbf{x}_t; \sqrt{1-\beta_t}\mathbf{x}_{t-1}, \beta_t I). \tag{27}$$

For an irreducible transition matrix $P$ with stationary distribution $\pi$, define:

$$\sigma_t(x, y) := \frac{P^t(x, y)}{\pi(y)}, \tag{28}$$

with $\sigma_t(x, y) = \sigma_t(y, x)$ when $P$ is reversible with respect to $\pi$. We also have:

$$< \sigma_t(x, \cdot), 1 >_\pi = \sum_y q_t(x, y)\pi(y) = 1. \tag{29}$$

Next, we have the definition of $l^p$-distance $d^{(p)}$ as:

$$d^{(p)}(t) := \max_{x \in \mathcal{X}} ||\sigma_t(x, \cdot) - 1||_p. \tag{30}$$

To replace the above notations with the notations from DDMs, we have:

$$d^{(1)}(t) := \max_{x \in \mathcal{X}} ||\sigma_t(x, y) - 1||_1, \tag{31}$$

and

$$\sigma_t(x, y) = \frac{P^t(x, y)}{\pi(y)} = \frac{x \sim \mathcal{N}(x_t; \sqrt{1-\beta_t}x_{t-1}, \beta_t I)}{y \sim \mathcal{N}(0, I_d)}. \tag{32}$$

Based on the definition of mixing time in 24, we then have:

$$t_{mix}^{(1)}(\varepsilon) := \inf\{t \geq 0; d^{(1)}(t) \leq \varepsilon\}. \tag{33}$$

Table 4: Gaussian radius estimation for empirical search of the mixing step for pre-trained DDPM on CelebA-64.

| Steps | 1000 | 900 | 800 | 700 | 600 | **500** | 400 | 300 |
|---|---|---|---|---|---|---|---|---|
| $\mathbf{x}_T^s + p_s$ | 110.84 | 110.82 | 110.83 | 110.58 | 109.83 | 107.85 | 103.22 | 94.64 |
| $\lvert \triangle r \rvert$ | 0.02 | 0.01 | 0.25 | 0.75 | 1.98 | **4.63** | 8.58 | - |
| $\mathbf{x}_T^s + p_i$ | 110.88 | 110.86 | 110.84 | 110.63 | 109.89 | 107.86 | 103.10 | 94.45 |
| $\lvert \triangle r \rvert$ | 0.02 | 0.02 | 0.21 | 0.74 | 2.03 | **4.76** | 8.65 | - |
| $\mathbf{x}_T^i + p_s$ | 95.06 | 93.30 | 91.56 | 90.14 | 88.69 | 86.61 | 82.80 | 76.46 |
| $\lvert \triangle r \rvert$ | 1.76 | 1.74 | 1.42 | 1.45 | 2.08 | **3.81** | 6.36 | - |
| $\mathbf{x}_T^i + p_i$ | 95.06 | 93.34 | 91.61 | 90.16 | 88.75 | 86.57 | 82.87 | 76.53 |
| $\lvert \triangle r \rvert$ | 1.72 | 1.73 | 1.45 | 1.41 | 2.18 | **3.70** | 6.34 | - |

We take the value $\varepsilon$ to be $\frac{1}{2}$, and thus arrive at:

$$t_{mix}^{(1)} := \inf\{t \geq 0; d^{(1)}(t) \leq \frac{1}{2}\}. \tag{34}$$

Now, we return back to Equation 32 and replace the Equation 34 with:

$$\max_{x \in \mathcal{X}} \lVert \frac{x \sim \mathcal{N}(x_t; \sqrt{1 - \beta_t} x_{t-1}, \beta_t I)}{y \sim \mathcal{N}(0, I_d)} - 1 \rVert \leq \frac{1}{2}. \tag{35}$$

By using the Proposition C.2, we can now substitute the above Equation 35 using the approximation as follows:

$$\lVert x \sim \mathcal{N}(x_t; \sqrt{1 - \beta_t} x_{t-1}, \beta_t I) \rVert \leq 4. \tag{36}$$

This above gives an approximation of the transition kernel at the mixing step in the diffusion direction we would expect, with a radius change at approximately 4.

We observe that there is no explicit dependency on the dimensionality of the latent spaces, but directly related to the formulation of the transition kernel, which is mostly the Gaussian as used in existing DDMs implementations. In the meanwhile, we note the intermediate latent encodings $x_t$ are actually dataset dependent. Therefore, we verify our claim that the mixing step is more dependent on the transition kernel and dataset. However, despite no explicit dependency between the mixing step and dimensionality, we empirically observe that the appearance of mixing step still differs in pre-trained diffusion models on different resolutions as in Tab. 5.

$\square$

Interestingly, the above proof gives us a numerical approximation for the transition kernel when the mixing step appears, which is the radius variation at approximately 4. We hereby finish demonstrating the fact that the mixing step appears at around diffusion step $t = 500$ in our main paper. In the meanwhile, other related works [31, 35] report similar conclusions that editing on step 500 shows better empirical performance in different experimental settings.

### D.2 More Empirical Results on Mixing Step

For the empirical verification of the mixing step, we use a pre-trained DDPM model [21] on the CelebA dataset [39] with $3 \times 64 \times 64$ resolution. Therefore, for a standard Gaussian space in the dimensionality of $d = 3 \times 64 \times 64 = 12,288$, the expected Gaussian radius is $r = \sigma\sqrt{d} = 110.85$. The full radius estimation results are listed in Tab. 4, we also show the difference in Gaussian radius between consecutive 100 steps. Note we are slightly "abusing" the estimation results for steps after the mixing step, since the distributions of the latent spaces after $\epsilon_{t_m}$ are no longer considered as Gaussian, but rather converge to the actual data distributions in $\mathcal{X}$, therefore, estimating the Gaussian radius of those latent spaces are not theoretically sound. This also explains the reason why we do not report the numbers for step numbers less than 300. The above also explains our design to derive the theoretical proof in the diffusion direction, but proposes to empirically search for the mixing step via the denoising direction.

Table 5: More results on Gaussian radius estimation for the empirical search of the mixing step for pre-trained DDMs

| Model | Setting | 1000 | 900 | 800 | 700 | **600** | **500** | 400 |
|---|---|---|---|---|---|---|---|---|
| DDPM-CelebA-HQ-256 | $\mathbf{x}_T^s + p_s$ | 443.42 | 443.36 | 443.30 | 442.49 | 440.16 | 432.55 | 416.77 |
| | $\lvert \triangle r \rvert$ | 0.06 | 0.06 | 0.81 | 2.33 | 7.61 | 15.78 | - |
| DDPM-CelebA-HQ-256 | $\mathbf{x}_T^s + p_i$ | 443.40 | 443.29 | 443.06 | 442.22 | 439.44 | 431.66 | 413.96 |
| | $\lvert \triangle r \rvert$ | 0.11 | 0.23 | 0.84 | 2.78 | 7.78 | 17.70 | - |
| iDDPM+AFHQ-256 | $\mathbf{x}_T^s + p_i$ | 443.34 | 443.21 | 442.84 | 441.72 | 439.31 | 429.16 | 408.69 |
| | $\lvert \triangle r \rvert$ | 0.13 | 0.37 | 1.12 | 2.41 | 10.15 | 20.47 | - |

## D.3 Connection with Existing Works

We notice that existing SOTA methods [31, 35] have proposed similar ideas in their works, by empirically exploring the diffusion steps that obtain better qualitative results. However, the mixing step has been studied as a hyper-parameter (*i.e.*, "return step" in [31] and "edit step" [35]) that influences the downstream qualities without formal definition. In this work, we formally define and introduce the concept of the mixing step, which originated from sound mathematical studies on the Markov mixing time, and provide a comprehensive perspective to re-think this "hyper-parameter". More excitingly, we discover that the theoretically driven deviation and our Gaussian radius estimation method come to a consistent conclusion and echo with previous literature in actual experimental tests.

## E  Boundary Search Discussion

In this section, we present more details about our proposed boundary search method, and discuss the connections between different latent space levels from the perspective of the Projection theorem.

### E.1  Projection Theorem

In theory, we expect the projected lower-dimensional subspace to preserve the same properties of its original higher-dimension space such as the projected distances between pairs of samples should have the same ordering in two spaces. In mathematics, we can ensure the validity of this projection design using the existing projection theorems.

*Theorem* E.1. **Theorem of the Random Projection.** *Let* $\mathbf{v}$ *be a fixed unit length vector in a $d$-dimensional space and let $W$ be a random $k$-dimensional subspace. Let* $\mathbf{w}$ *be the projection of* $\mathbf{v}$ *onto $W$. For any* $0 \le \epsilon \le 1$, $Prob(\lvert \lVert \mathbf{w} \rvert^w - \frac{k}{d} \rvert \ge \epsilon \frac{k}{d}) \le 4e^{-\frac{k\epsilon^2}{64}}$.

One way to interpret the random projection theorem is that if one chooses a random $k$-dimensional subspace from a higher-dimensional space in $d$-dimension, then indeed all the projected distances in the $k$-dimensional space are approximately within a known scale factor of the distances in the $d$-dimensional space.

We present the projection theorem here to draw connections between the above boundary search and different operational latent space levels (*i.e.*, $\epsilon$-space and $h$-space). As we describe in the main paper, the classification results from Tab. 6 show that even though the accuracy score is generally lower in $\epsilon$-space, it does carry meaningful semantic boundaries. This above observation and claim differ from the previous literature [44], where the latent spaces of DDMs are considered to lack semantic meaning. In fact, given the recent study from [35], which first reveals the semantic behaviors of pre-trained DDMs in $h$-space, it provides evidence to imply that the same semantic meanings might also exist in the higher-dimensional $\epsilon$-space. As $h$-space is a subspace of corresponding $\epsilon$-space with higher dimensionality.

## F  *BoundaryDiffusion* with Mixing Trajectory

We show the algorithm implementation for our proposed *BoundaryDiffusion* with mixing trajectory under the conditional application scenario in Algo. 1. For the unconditional scenario, the only difference is that we can directly sample the latent encodings from the Gaussian distribution as the initial $\mathbf{x}_T$, and get the corresponding $h$-level latent encoding $\mathbf{h}_T$ from the given DDPM at $T$ step.

---

**Algorithm 1** BoundaryDiffusion (Conditional)

---

**Input:** input image $\mathbf{x}_0$, target boundaries $\mathbf{b}_\epsilon$ and $\mathbf{b}_h$ for the editing attribute $m$, pre-trained DDM $p$, inversion steps $S_{inv}$, denoising steps $S_{gen}$, mixing step $t_m$, user defined editing distance $\zeta_\epsilon$ and $\zeta_h$, and editing space steps $K$.

// *Step 1: Inversion via DDIMs to get the latent encoding at $t_m$*

Define $\{\tau_s\}_{s=1}^{S_{inv}}$ s.t. $\tau_1 = 0, \tau_{S_{inv}} = t_m$

**for** $s = 1, 2, ..., S_{inv} - 1$ **do**
    $\epsilon \leftarrow p(\mathbf{x}_{\tau_s}, \tau_s)$
    $\mathbf{x}_{\tau_{s+1}} = \sqrt{\alpha_{\tau_s}}\mathbf{x}_{\tau_s} + \sqrt{1 - \alpha_{\tau_s}}\epsilon$
**end for**

$\mathbf{h}_{t_m} \leftarrow$ extract $h$ feature map from $\epsilon$

// *Step 2: Boundary guidance*

// *Step 2.1: Define initial editing space in $\epsilon$ and $h$ latent levels*

$\{d_\epsilon^j\}^K$ s.t. $d_\epsilon^1 = -\zeta_\epsilon, d_\epsilon^K = \zeta_\epsilon$

$\{d_h^j\}^K$ s.t. $d_h^1 = -\zeta_h, d_h^K = \zeta_h$

// *Step 2.2: Compute projection distance to the boundaries*

$\{d_{p,\epsilon}\} = \{d_\epsilon\} - \mathbf{b}_\epsilon^T \mathbf{x}_{t_m}$

$\{d_{p,h}\} = \{d_h\} - \mathbf{b}_h^T \mathbf{h}_{t_m}$

// *Step3: Denoising with mixing trajectory*

**for** k = 1, 2, ..., K **do**
    $\mathbf{x}'_{t_m} = \mathbf{x}_{t_m} + d_\epsilon^k \mathbf{b}_\epsilon$
    $\mathbf{h}'_{t_m} = \mathbf{h}_{t_m} + d_h^k \mathbf{b}_h$
    $\mathbf{x}_s \leftarrow \mathbf{x}'_{t_m}$
    $\mathbf{h}_s \leftarrow \mathbf{h}'_{t_m}$
    **for** $s = S_{gen}, S_{gen} - 1, ..., 2$ **do**
        $\epsilon \leftarrow p_s(\mathbf{x}_s, \mathbf{h}_s, s)$
        $z \sim \mathcal{N}(0, I_d)$
        $\mathbf{x}_{s-1} = \sqrt{\alpha_{s-1}}(\frac{\mathbf{x}_s - \sqrt{1-\alpha_s}\epsilon}{\sqrt{\alpha_s}}) + \sqrt{1 - \alpha_{s-1} - \sigma^2}\epsilon + \sigma_s z$
    **end for**
**end for**

---

# G  More Experimental Results

We present more experimental results and discussion in this section.

## G.1  Pre-trained Models and Datasets

The pre-trained DDMs we use for experiments mainly include the DDPM [21] and the improved DDPM (iDDPM) [41]. The main difference between the original DDPM and the improved version lies within the fact that iDDPMs use a hybrid learning objective that obtains better log-likelihoods than directly optimizing it.

We conduct experiments on multiple datasets, which includes CelabA-64 [39], CelebA-HQ-256 [27], AFHQ-dog-256 [10], LSUN-church-256 [66], LSUN-bedroom-256 [66]. Different from existing works that usually pay little attention to the image resolutions in the experiments, the resolutions play an important role in our experiments since they define the actual dimensionality of the latent spaces for pre-trained DDMs. However, the $64^2$ resolution model is mainly used in the high-dimensional analysis and interpolation observations, for the image editing and semantic control experiments, we use $256^2$ as the default resolution for visualization quality.

## G.2  Semantic Boundary Validation

We search the semantic boundaries via linear classifiers on both $\epsilon$-space and $h$-space using 100 images, and we show the semantic behaviors via the testing classification accuracy on different attributes in Tab. 6. We observe from the classification results that the boundaries in $h$-space are in general better defined compared to the $\epsilon$-space, which is consistent with previous findings from [35]. Notably, for certain attributes such as *glass* and *mustache*, both space levels perform well in defining the

boundaries, which implies and aligns with our empirical finding that guidance on both levels of latent spaces helps for more effective semantic control.

Table 6: Classification accuracy on separation boundaries in different latent spaces at the mixing step.

| Latent space | Smile | Glass | Age | Mustache |
|:---:|:---:|:---:|:---:|:---:|
| $\epsilon$ | 0.86 | 0.95 | 0.87 | 0.96 |
| $h$ | **0.98** | 0.95 | **0.93** | 0.96 |

### G.3  User Study

As subjective evaluations, we conduct a user study to compare our proposed method with Asyrp [35] and DiffusionCLIP [31] on CelebA-HQ-256 [39]. We use the official codebases from previous works and follow the exact default commands, using the *smile* attribute as the editing target, either to add or remove smiles from 100 raw images that are randomly selected from the dataset. Our human evaluators include 20 graduate students volunteers, with 15 males and 5 females. Specifically, we show the original images together with edited images to the human evaluator and ask them to pick the best result with respect to general quality and attribute editing effectiveness.

More **non-cherry picky** editing results from three different methods are included in Fig. 5 and Fig. 6.

Fig. 7 shows more interpolation results from different combinations of latent encoding sources and sampling methods on CelebA-64 during our investigation on the distance effect. More qualitative samples for different editing settings are included in Fig. 8, Fig. 9, Fig. 10 and Fig. 11.

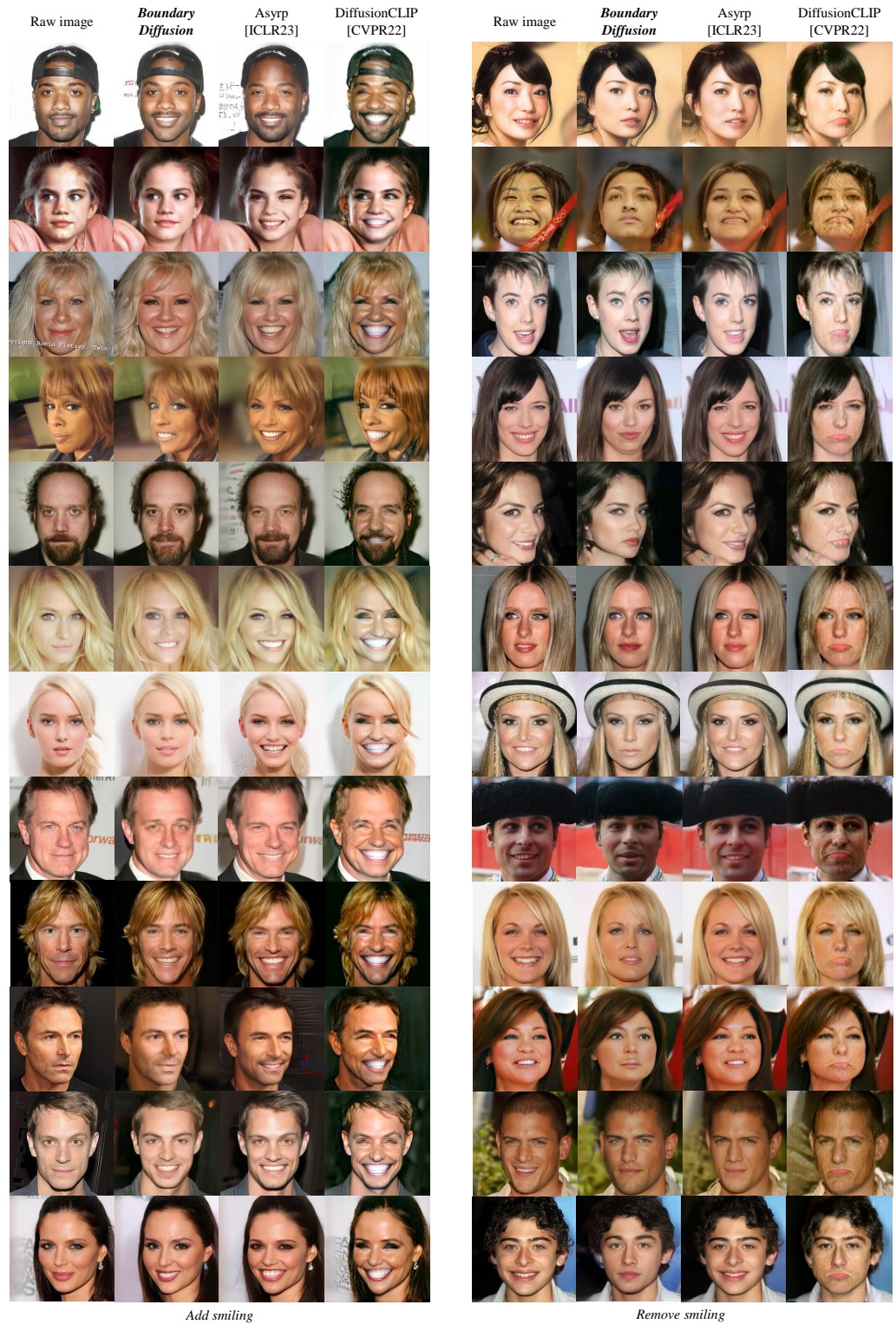

Figure 5: **Non-Cherry-Picked** randomly selected results for *add smiling* and *remove smiling* editing operations from our proposed *BoundaryDiffusion*, Asyrp [35], and DiffusionCLIP [31].

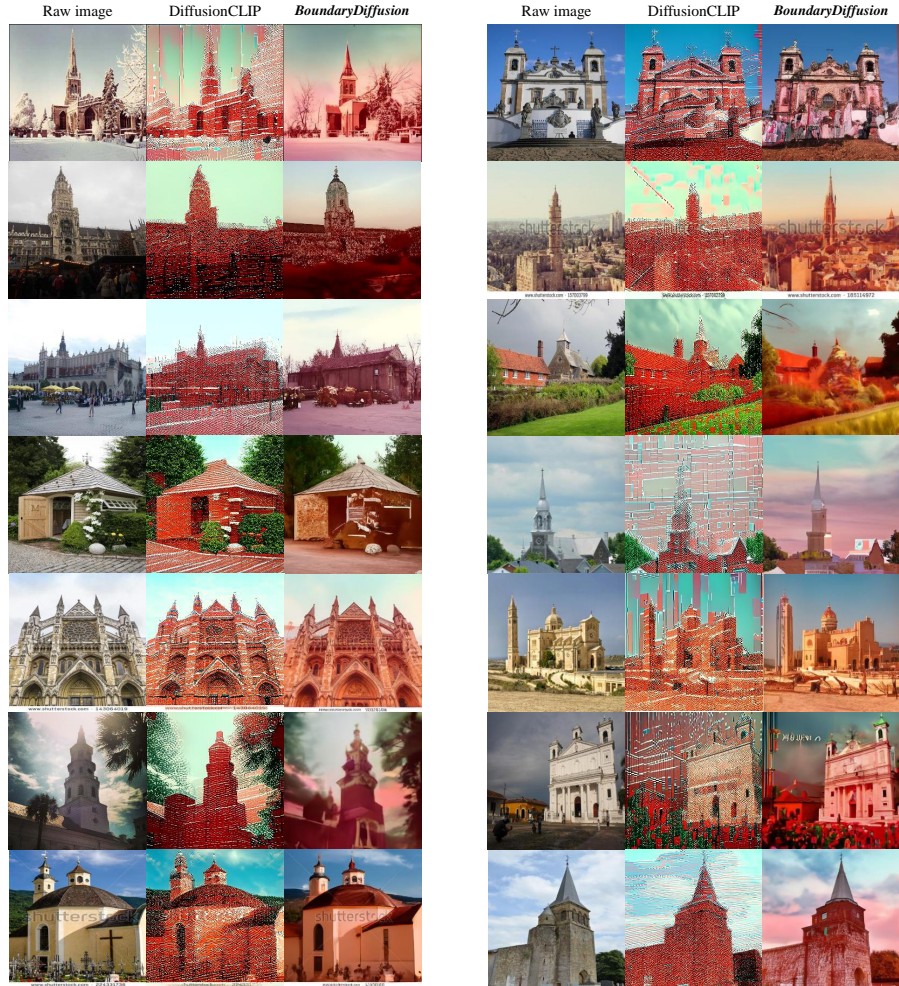

Figure 6: **More Non-Cherry-Picked** randomly selected results for LSUN-Church dataset given *"red brick church"* as editing text prompt. Comparing against learning-based SOTA method DiffusionCLIP, our learning-free method preserves better image details and achieve better fidelity and more natural editing effects.

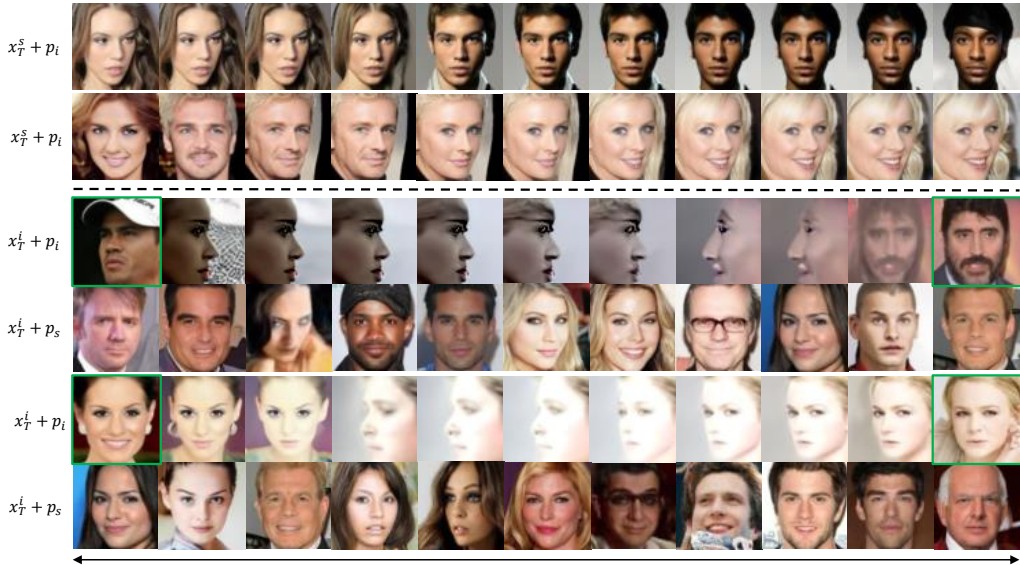

Figure 7: More interpolation results from different combinations of latent encoding sources and sampling methods on CelebA-64.

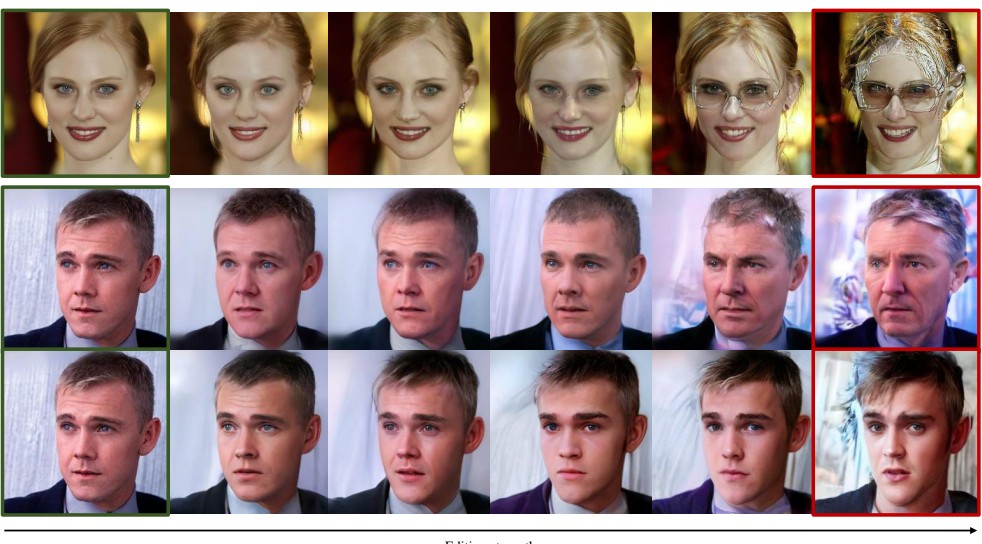

Editing strength

Figure 8: More qualitative results for editing strength modification. We use the CelabA-HQ-256 dataset and the attributes *glass* and *age* as examples. In particular, we also show samples with distortions and lower quality when the editing distance becomes too large. The optimal editing distance range is also related to the properties of the high-dimensional spaces.

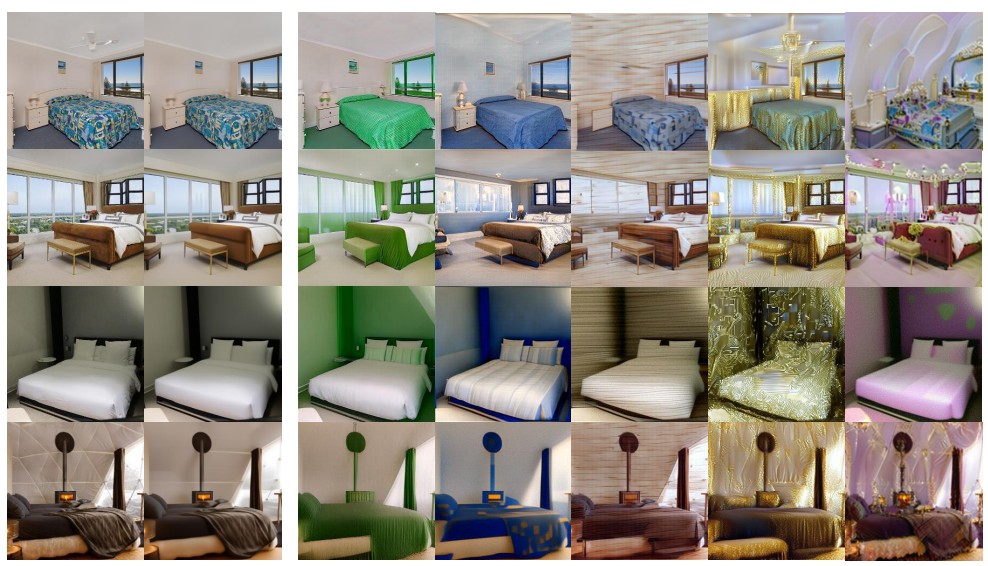

Figure 9: More qualitative results for text-based conditional editing on the LSUN-Bedroom-256 dataset.

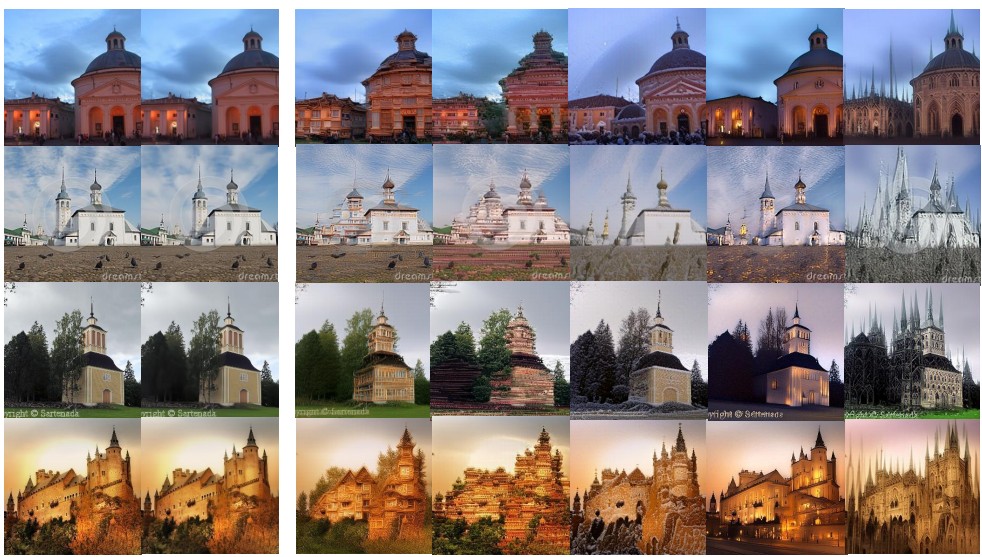

Figure 10: More qualitative results for text-based conditional editing on the LSUN-Church-256 dataset.

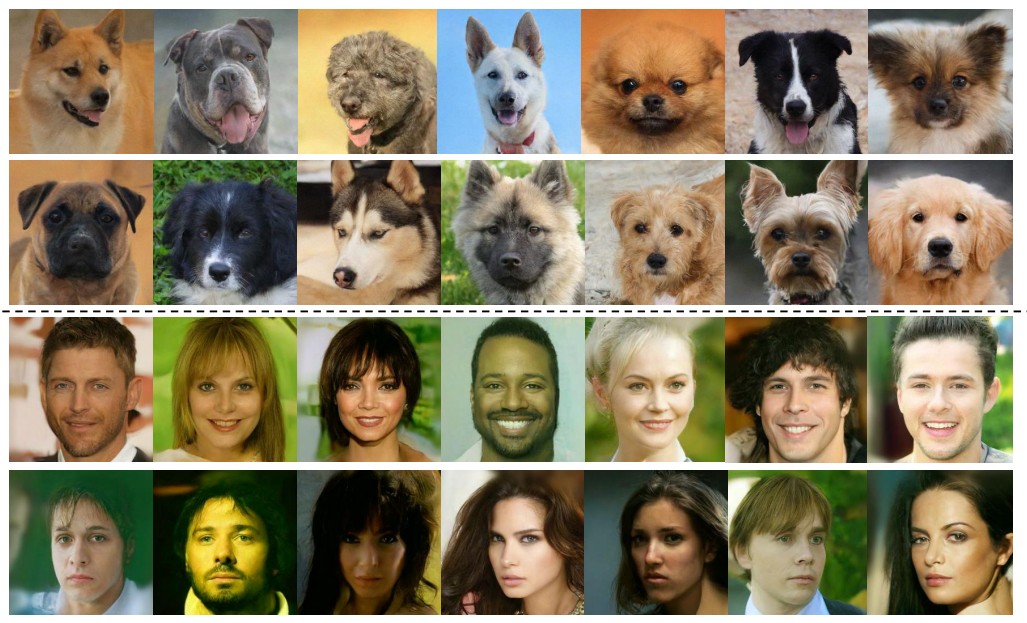

Figure 11: More qualitative results for unconditional semantic control on AFHQ-Dog-256 and CelabA-HQ-256 datasets with *smile* semantic.

