# OpenReview forum: "Boundary Guided Learning-Free Semantic Control with Diffusion Models"
_NeurIPS.cc/2023/Conference — NeurIPS 2023 poster_

### Official Review · Reviewer_TWGA · 2023-06-19

**Soundness:** 2 fair
**Presentation:** 2 fair
**Contribution:** 3 good
**Rating:** 5
**Confidence:** 3

**Summary:**

This paper introduces the BoundaryDiffusion method for semantic image editing with frozen pre-trained Denoising Diffusion Models. The authors provide an analysis showing that the inverted latent variables have a different distribution from the prior Gaussian. From this observation, this paper suggests a method for identifying the critical step for semantic editing. This paper proposes a method for discovering semantic control perturbations through the linear SVM classifier of the corresponding semantics on this critical step.

**Strengths:**

- This work is overall well-written.
- This work showed that $\mathcal{X}$ also provides a semantic property.
- The analysis in Sec 3 is interesting as it shows the difference between the inverted latent variables and the Gaussian samples.

**Weaknesses:**

- The BoundaryDiffusion does not require training a neural network. But, it requires the linear SVM classifier optimized from the annotated images.
- The discussions of Distance effect in Sec 3 requires some clarifications (Please see my Questions below.).

**Questions:**

- **Q1-2. (L175-177)** The homogeneity of latent space in [1] was mainly about the semantic property of $h$-space. However, L175-177 discusses the spatial property (radius) of the inverted latent variable $x_{T}^{i}$ and the sampled latent variable $x_{T}^{s}$. Can we discuss the spatial property of $\mathcal{X}$-space from the semantic homogeneity of $h$-space? If so, how can we understand the mixing step under the maintained spatial difference after the deterministic denoising $p_{i}$? Aren’t the inverted latent variables and sampled latent variables supposed to be close enough in the mixing step?

- **Q3-4. (L180)** Lemma B.2 states that the volume of a *unit-radius* sphere decreases to near-zero as the dimensionality grows. However, the mass of high-dimensional Gaussian concentrates around the radius of $r=\sigma\sqrt{d}$. In this case, it appears that the volume of sphere, i.e., $V(d) r^{d}$ does not converge to zero when approximated with Stirling's formula. It would be beneficial if the authors could provide further clarification on this matter to enhance our understanding. Moreover, if the volume decreases to zero, isn’t it harder for the stochastic denoising to stay in the higher concentration mass (**L186-188**)?

[1] Mingi Kwon, Jaeseok Jeong, and Youngjung Uh. Diffusion models already have a semantic latent space. In ICLR, 2023.

**Limitations:**

The authors addressed the limitations and potential negative societal impact of their work.

---

> ### Author Rebuttal · Authors · 2023-08-04
>
> We deeply appreciate the feedback regarding the theoretical part of the distance effect.
>
> ---
> **Comment 1:** *BoundaryDiffusion* does not require training a neural network but requires the linear SVM classifier optimized from the annotated images.
>
> **Response to Comment 1:**
>
> &nbsp;&nbsp;&nbsp; We are more of **searching** and **fitting** the existing boundaries with SVM as hyperplanes, rather than **learning classifiers** since we are exploring the **intrinsic semantic properties** of pre-trained DMs formed in training. Note this high-level take-away is critical since we proved that **an unconditionally probabilistic generative model learns meaningful semantics by itself**. This is **not evident** in previous literature where probabilistic generative models, including DMs, are conventionally believed to be not good at learning data semantics **without explicit semantic supervisions** such as text.  Also, our *BoundaryDiffusion* is much more computationally economical than learning-based SOTA [1, 2] with negligible fitting time. As for annotated images, [1,2] also require annotated images for fine-tuning or learning the editing neural networks.
>
> ---
> **Questions on the distance effect:**
>
> - **Q1:** The homogeneity of latent space in [1] was mainly about the semantic property of $h$-space. L175-177 discusses the spatial property (radius) of the inverted and the sampled latent variable. Can we discuss the spatial property from the semantic homogeneity of $h$-space?
> - **Q2:**  If so, how can we understand the mixing step under the maintained spatial difference after deterministic denoising $p_i$? Aren’t the inverted and sampled latent variables supposed to be close enough in the mixing step?
> - **Q3:** Lemma B.2 states that the volume of a unit-radius sphere decreases to near-zero as dim grows. However, the mass of high-dimensional Gaussian concentrates around the radius $r$. In this case, it appears that the volume of sphere does not converge to 0 when approximated with Stirling's formula.
> - **Q4:** If the volume decreases to 0, isn’t it harder for the stochastic $p_s$ to stay in the higher concentration mass?
>
> **Our responses:**
>
> &nbsp;&nbsp;&nbsp; Firstly, we would like to emphasize **an important precondition** for the studies on the distance effect: all the spatial/probabilistic properties and empirical analytical tools (*e.g.*, the mass concentration and radius) are built upon the **Gaussian distribution prior**. This is because an arbitrary high-dimensional distribution is extremely complex to parse and understand, while Gaussian has many well-established properties that help us to better interpret DMs. Therefore we consider the Gaussian prior as a **reference** to understand DMs, and restrict our studies mostly to the latent space of step $T$, which exhibits a **standard Gaussian by definition**.
>
> - **A1: We can discuss the semantic homogeneity but not the spatial property in $h$-space.** For the spatial property, it is unrealistic to conduct similar studies in $h$-space at step $T$, since $h$-space is extracted from the bottleneck layer of U-Net, and the Gaussian prior no longer holds after convolutional layers from U-Net, and thus we do not have a *reference or expectation* on what the spatial properties should be. However, we can indeed conduct the semantic homogeneity analysis in $h$-space using the SVM classifier as the tool and found that the semantic properties are preserved in $h$-space, as demonstrated in **Appendix G.2, Tab. 3**.
>
> - **A2: Distance effect is observed for inverted latent variables after distance shift following deterministic denoising; the mixing step is an intrinsic property formed during training to characterize the formation time of meaningful semantics.**
> To observe the distance effect, there are two preconditions based on our experience: the departure latent encoding locates in a non high concentration area, and the denoising process does not bring it back along the trajectory. However, the mixing step is a slightly different concept with distance effect. Our high-level take-away for the mixing step is that this characterizes the time when the semantics start to form in a separable manner along the denoising trajectory, which is an intrinsic property formed in training given a pre-trained model.
>
> - **A3:  Please see the clarifications below.**
>
>    *i)* It is true that the mass of high-dimensional Gaussian concentrates around the $r$, but **within an annulus** of width $1/d$, as shown in right of Fig.2(b). Therefore, when approximating with Lemma B.2, we need to take the width into consideration.
>
>    *ii)* A simple way to check the formula from Lemma B.2: we have $A(2)=2\pi$, $V(2)=\pi$ and $A(3)=\frac{2\pi^{3/2}}{1/2\sqrt{\pi}}=4\pi$, $V(3)=\frac{2}{3}\frac{\pi^{3/2}}{\Gamma(3/2)}=\frac{4}{3}\pi$. Note that $\pi^{3/2}$ is an exponential in $\frac{d}{2}$, and $\Gamma(\frac{d}{2})$ grows as the factorial of $\frac{d}{2}$, which implies that $lim_{d\to\infty }V(d)=0$.
>
>    *iii)* In addition, essentially, we know the volume of concentration mass goes to zero for **an arbitrary high-dimensional distribution** according to the **curse of dimensionality**.
>
> - **A4: Yes and no.**
>
>    *i)* Yes, it is always harder to train deep learning models with data in higher dimensionality **in general**, which is also part of the curse of dimensionality.
>
>    *ii)* No because the DMs are trained with variational loss with their max probabilistic nature. If we are comparing $p_s$ and $p_i$, which are both **sampling process** that does not touch the training, $p_s$ is more consistent with training.  For DDIM $p_i$, the training is identical to DDPM $p_s$, and bears a slight quality decrease as explicitly discussed in its original paper [3].
>
> ---
> [1] Diffusion models already have a semantic latent space. In ICLR, 2023.
>
> [2] Diffusionclip: Text-guided diffusion models for robust image manipulation." In CVPR, 2022.
>
> [3] Denoising Diffusion Implicit Models. In ICLR, 2021.

---

> > ### Comment · Reviewer_TWGA · 2023-08-14
> >
> > Thank you for the response. I have a few additional follow-up questions.
> >
> > - **Q3.** I misunderstood that the concentration of mass in Line 180 refers to the sphere of radius $r=\sigma \sqrt{d}$. Then, what does $O(\frac{r}{\sqrt{d}})$ represent in Fig 2(b)? The width or volume? Could you provide a reference for $O(\frac{r}{\sqrt{d}})$?
> >
> >    - (Added) I found out that $O(\frac{r}{\sqrt{d}})$ denotes the width of the slice for the concentration mass from the rebuttal for Reviewer sSmU. If I understood correctly, $r=\sigma \sqrt{d}$. It is hard to understand that the volume of the slice of width $O(1)$ (if we assume $\sigma=1$) decreases to zero as the dimensionality grows. Could you provide further clarification? Lemma B.4. only states about the fraction of the volume.
> >
> > - **Q1.** I would like to clarify my initial questions. I agree with the authors that we can only discuss the semantic homogeneity in $h$-space. Actually, that was what I wanted to ask. My initial question was about understanding Line 175-177. I interpreted this sentence as discussing "spatial difference" based on the homogeneity of $h$-space. Could you provide further clarification?
> >
> > > This spatial difference remains after the deterministic denoising via $p_i$ due to the homogeneity of intermediate latent spaces [22], resulting in denoised samples from $x_{T}^{i}$ to be rather out-of-distribution and low-fidelity after distance shift in latent spaces.

---

> > > ### Author Response · Authors · 2023-08-14
> > > **Further Clarifications**
> > >
> > > Thank you for reading our rebuttal and for the further questions! We are very glad to further clarify the questions below.
> > >
> > > ---
> > > - **Q1:** I interpreted this sentence as discussing "spatial difference" based on the homogeneity of $h$-space. Could you provide further clarification?
> > >
> > > - **A1:**  Thank you for the question, we believe it is very insightful to understand the latent spaces of diffusion models in terms of different space levels.
> > >
> > >    To make a very clear answer to this question, we would like to distinguish between two subtitle types of spatial difference here: 1. **Quantitative spatial difference** measurable by the Gaussian metrics (unique in our work) 2. **Qualitative spatial difference** characterizable by the semantics (not explicitly proposed in [1], but can be reasonably interpreted this way). To be more specific, the first type of spatial difference is more strict and requires the Gaussian prior as we have mentioned in our initial rebuttal, which is also the spatial difference we refer to in the context of distance effect analysis. In comparison, the second type of spatial difference is more qualitative and relaxed, which can also be understood as the ``relative spatial locations to the invariant semantic boundaries”. Since the second type of spatial difference is more qualitative and does not require Gaussian prior, it can also be applicable in $h$-space, as long as the semantic boundaries **exist and stay invariant** along the denoising process. In other words, case 1 should imply case 2, but not the contrary.
> > >
> > >    Now back to the question, given specific context, the spatial difference shall refer to different definitions. In our initial rebuttal, given the context of distance effect analysis, we position ourselves in the first case, where the spatial distance is measurable and quantified by the Gaussian metrics. Given the context of homogeneity of semantics in $h$-space [1], we are in the second case, where the Gaussian prior does not hold, but the semantic properties are still preserved. Therefore, "out-of-distribution" is more of a qualitative property applicable to the relaxed definition of spatial difference.
> > >
> > > ---
> > > - **Q3:** Could you provide further clarification on the zero volume as the dimensionality grows?
> > >
> > > - **A3:** For sure. We agree that sometimes the behaviors of high-dimensional distributions are not at all intuitive, and there are usually **various angles** to approach the same conclusion. Among different angles/ways of explanations, we find the following one the most straightforward to understand the zero-volume conclusion, but other angles of explanations and rigorous proofs can also be found in [4].
> > >
> > >    Given the condition of a **standard Gaussian** with $\mu = 0$, $\sigma = 1$ and dimensionality $d$, we know that it geometrically presents a **unit sphere**. And in mathematics, the ratio of the volume of a sphere of radius $1-\varepsilon$ to the volume of a unit sphere in $d$-dimensions is: $\frac{(1-\varepsilon)^dV(d)}{V(d)} = (1 - \varepsilon)^d$, thus goes to 0 as $d$ goes to infinity, when $\varepsilon$ is a fixed constant. We only need to further verify the condition of *$\varepsilon$ is a fixed constant*, which is luckily true in our case given the fact that we have the width of annulus $O(\frac{r}{d})$. ( As $d$ grows, the width of the slice $O(\frac{r}{\sqrt{d}})$ does not impose direct influence to the zero-volume conclusion based on the above proof, we introduced this with the hope to give a more vivid visual illustration of the probability concentration mass in Fig. 2. We will further add the meaning of $O(\frac{r}{\sqrt{d}})$ in the figure caption to avoid potential confusions in our final manuscript. )
> > >
> > > ---
> > > - **Additional note from authors:** In addition, we thought the reviewer might also be interested in our new findings: We have confirmed in our follow-up additional experiments that the proposed *BoundaryDiffusion* is also **applicable to LDM (StableDiffusion)**, meaning that the latent image representation from encoder-decoder also preserves the invariant semantics we have proposed in this work. We will release the codes and integrate semantic boundaries we have located into the public platform (*e.g.*, HuggingFace) to allow easy access for the research community.
> > >
> > >
> > > ---
> > > [4] Foundations of data science. Cambridge University Press, 2020.

---

> > > > ### Comment · Reviewer_TWGA · 2023-08-16
> > > >
> > > > I appreciate the authors for their efforts for further clarifications. However, in the $d$-dimensional Gaussian, the probability mass is concentrated around the sphere of radius $r=\sigma \sqrt{d}$, not the unit sphere. The volume of the ball with a radius of $r=\sigma \sqrt{d}$ is $r^{d}$ times greater than that of the unit sphere. Hence, the dynamics are significantly different from that of the unit sphere, because the volume of the ball with a radius of $r=\sigma \sqrt{d}$ goes to infinity as $d$ goes to infinity.
> > > >
> > > > Whether the volume of concentration mass decreases to near-zero as the dimensionality grows (Line 180), I believe, is not a serious concern to the contribution of this paper. My concern is about ensuring the accuracy of the statement presented in this paper.

---

> > > > > ### Author Response · Authors · 2023-08-16
> > > > >
> > > > > We really appreciate the careful reading of the reviewer, the in-depth feedback, and the acknowledgment of our work.
> > > > >
> > > > > We are pretty sure about the correctness of the zero-volume conclusion for high-dimensional Gaussian, as this has been an established property in mathematics. But at the same time, we will come up with a more clear theoretical proof adapted to our context for readers to understand in an easier way, and provide them in our final paper.

---

> > > > > > ### Comment · Reviewer_TWGA · 2023-08-17
> > > > > >
> > > > > > Please let me know if there are any errors in the calculations provided below.
> > > > > >
> > > > > > ----
> > > > > >
> > > > > > Assume $\sigma=1$, then the radius of concentration of mass $r=\sigma \sqrt{d}=\sqrt{d}$. Let $V_{r} (d)$ be a volume of $r=\sqrt{d}$ ball.
> > > > > >
> > > > > > $$
> > > > > > V_{r} (d)=r^{d} \cdot V(d) = \frac{\pi^{d/2} \cdot \sqrt{d}^{d}}{(2/d) \cdot (2/d)!}
> > > > > > $$
> > > > > > The definition of $V(d)$ is from Lemma B.2 in the appendix. Then, by Stirling's formula,
> > > > > > $$
> > > > > > V_{r} (d) \approx \frac{\pi^{d/2} \cdot \sqrt{d}^{d}}{(2/d) \cdot \sqrt{2 \pi (d/2)} \cdot (\frac{d}{2e})^{d/2}}
> > > > > > = \frac{\pi^{d/2} \cdot (2e)^{d/2}}{(2/d) \cdot \sqrt{ \pi d} } \rightarrow \infty \hspace{5pt} \text{ as $d$ goes to } \infty.
> > > > > > $$
> > > > > > The last conclusion holds because the numerator increases exponentially.
> > > > > >
> > > > > > Now, let's consider the volume of the interior of the annulus of width $\frac{r}{d}$, i.e., the ball of radius $r_{0}=r(1-\frac{1}{d})$.
> > > > > > $$
> > > > > > V_{r_0} (d)=(1-\frac{1}{d})^{d} \cdot V_{r}(d).
> > > > > > $$
> > > > > > Note that $\lim_{d \rightarrow \infty} (1-\frac{1}{d})^{d} = \frac{1}{e}$. Therefore, $V_{r_0}(d)$ only contains $\frac{1}{e}$ of $V_{r}(d)$ where $V_{r}(d)$ goes to infinity as $d$ goes to infinity.
> > > > > >
> > > > > > ---
> > > > > > Then, how do we arrive at the conclusion that the volume of mass concentration, i.e., $V_{r}(d)-V_{r_0}(d)$, goes to zero as $d$ goes to infinity?

---

> > > > > > > ### Author Response · Authors · 2023-08-17
> > > > > > >
> > > > > > > We really appreciate the efforts of the reviewer! After revisiting our entire manuscript about the high-dimensional Gaussian part, the original book, as well as the reviewer's initial comments and provided calculations, we make further clarifications below. (Apologize that it seems that we have some misunderstanding on the definitions of the term *volume of concentration mass of high dimensional Gaussian* in previous discussions.)
> > > > > > >
> > > > > > > - **Clarification 1:** The conclusion from Lemma B.2 is correct, which states that the volume of a unit-sphere goes to zero as dimensionality grows, and holds for the case of any sphere with a fixed radius. (also acknowledged by the reviewer if we understand correctly)
> > > > > > >
> > > > > > > - **Clarification 2:** What is exactly the definition of "volume of concentration mass of high-dimensional Gaussian", and "goes to zero as dimensionality grows" ? (this seems to be where the misunderstanding begins)
> > > > > > >
> > > > > > >    - When we first referred to this in our manuscript, we borrowed the concept from existing machine learning literature such as [5,6], where the "volume of concentration mass of arbitrary high-dimensional distributions tend to go zero as the dimensionality grows", in a sense that *as the dimensionality of a space increases, a larger portion of the data points tend to cluster around specific areas of the space, leaving the rest of the space almost empty*. This corresponds to the case presented in our paper, where the samples are essentially contained in a thin and narrow slice of annulus. In this case, the existing literature understands the term in the sense of **"the portion of the volume where the concentration locates compared to the actual space volume goes to zero as dimensionality grows"**.
> > > > > > >
> > > > > > >    - Next, during the rebuttal when the reviewer raised the concern about the zero volume claim and provided the calculation (thank you again for the efforts), we realized the reviewer referred to the idea of the **"actual geometric volume of such a slice of the annulus in d-dimensional space"**, which also makes a lot of sense, please see our modified version for your provided calculation below.
> > > > > > >
> > > > > > >    - Afterward, we revisit the literature and found **there exists no formal exact definition for the term**. It seems most machine learning researchers tend to understand it as the first way, but we agree with the second way as pointed out by the reviewer establishes a more rigorous math meaning.
> > > > > > >
> > > > > > > - **Corrected Calculations:** In addition to the above clarifications, we also checked the calculation provided by the reviewer and found a few typos in the proof, and provided our version for your reference, but the idea is in general the same.
> > > > > > >
> > > > > > >    > According to the Lemma B.2, the volume of a unit-sphere is $V(d) = \frac{\pi^{d/2}}{\frac{d}{2}\Gamma(d/2)}$ *(not $2/d$, but $d/2$)*
> > > > > > >
> > > > > > >    > For a sphere of radius $r$, the volume is $V_r(d) = r^d V(d) = \frac{\pi^{d/2}\sqrt{d}^d}{\frac{d}{2}\Gamma(d/2)}$
> > > > > > >
> > > > > > >    > Based on Stirling's formula for the gamma function, we have $V_r(d) \approx \frac{\sqrt{d}^d \pi^{d/2}}{(d/2) \cdot \sqrt{2\pi(2/d)} \cdot (\frac{d}{2e})^{d/2}} = \frac{\pi^{d/2} \cdot (2e)^{d/2}}{d\sqrt{\pi d}}$
> > > > > > >
> > > > > > >    > The rest should be similar to the provided calculation.
> > > > > > >
> > > > > > >    Therefore, based on our discussions above, we believe the best way to interpret what we all want to convey here is: **One needs to increase the radius of the sphere to nearly $\sqrt{d}$ before there is a non-zero volume and hence significant probability mass due to the geometric properties of high-dimensional Gaussian samples.** This does not contradict with our initial responses to the last question, which believes stochastic denosing should be beneficial for a sampling process to stay in the higher probability area compared to deterministic ones.
> > > > > > >
> > > > > > > Hope we have clarified the question the reviewer has, and please let us know if there are any further concerns, thank you again!
> > > > > > >
> > > > > > > ---
> > > > > > > [5] The curse (s) of dimensionality. Nat Methods 15, no. 6 (2018): 399-400.
> > > > > > >
> > > > > > > [6] The curse of dimensionality in data mining and time series prediction., 2005.

---

> > > > > > > > ### Comment · Reviewer_TWGA · 2023-08-18
> > > > > > > >
> > > > > > > > Unfortunately, I do not agree with the authors that "most machine learning researchers tend to understand it as the first way". Also, I am unconvinced how the authors would correct this inaccurate statement. Hence, I would lower my initial evaluation to 4.

---

> > > > > > > > > ### Author Response · Authors · 2023-08-18
> > > > > > > > >
> > > > > > > > > Dear Reviewer,
> > > > > > > > >
> > > > > > > > > Thank you for your comments. We would like to further clarify the concerns you may have.
> > > > > > > > >
> > > > > > > > > - In the last reply, "It seems most machine learning researchers tend to understand it as the first way," is **gently based on the fact we found no formal definitions** for the term "volume of concentration mass" in this context (please kindly refer to references if you know any with a formal definition, it would also be helpful to our work and understanding), also on the fact that **"the curse of dimensionality"** seems to be relatively well-known in the machine learning area.
> > > > > > > > >
> > > > > > > > > - In terms of the correctness of the manuscript, the only phrase we mention the volume is in Line 180 in the main paper **"In addition, the volume of concentration mass decreases to near-zero as the dimensionality grows [1]"**, which is an additional information we provided given the context of **a high-dimensional sphere**, which is not false in case of fixed radius but we agree the concept may be not well-defined, but this statement also **stands alone from the remaining part of the paper for its own information purpose**.
> > > > > > > > >
> > > > > > > > >    This is also the statement which we want to prove during the rebuttal discussion, but later realized that the reviewer refer to the volume of a sphere with a changing radius, as we have clarified in our previous reply.
> > > > > > > > >
> > > > > > > > >    **Also given our context with a pre-trained diffusion model**, the dimensionality (which corresponds to the resolutions of images, and we have also experimented with 64x, 256x) is fixed **so that the expected radius is fixed**, as shown in Tab. 1 in the main paper and Tab. 1 & 2 in appendix.
> > > > > > > > >
> > > > > > > > > - In terms of further modification and clarifications for future readers, we agree that the most accurate state could be: *One needs to increase the radius of the sphere to $\sqrt{d}$ nearly before there is a non-zero volume and hence significant probability mass due to the geometric properties of high-dimensional Gaussian samples.* This is actually **exactly what has supported our claim and empirical finding** to explain why stochastic denosing is better than deterministic ones, due to the fact that they maintain **a larger radius** closer to the concentration mass, and thus supportive to our claim that "stochasticity helps with diffusion generation" in Line 183 - 188, also **clearly answer the initial Q4**, that stochastic process is easier to stay in the high concentration mass area.
> > > > > > > > >
> > > > > > > > >    Therefore, changing the sentence from Line 180 **"In addition, the volume of concentration mass decreases to near-zero as the dimensionality grows [1]"** to the current statement seems to be the best way to clarify, and also an easy modification that does not influence the remaining part of the paper.
> > > > > > > > >
> > > > > > > > > - Lastly, we kindly but also urgently ask the reviewer to think as the project as a whole, in which we have cautiously **introduced this novel perspective** to better understand the diffusion models, provided **abundant analytical empirical demonstrations**, proposed **a very effective and unified method framework**, and demonstrate its effectiveness with **SOTA performance in various task settings and model variants (DDPMs, iDDPMs, LDMs)**. As as mentioned many times by the reviewer and others, we believe the contributions of this work should worth the credits and should be beneficial for the research community.

---

> > > > > > > > > > ### Author Response · Authors · 2023-08-18
> > > > > > > > > > **Summarization for the Discussions on the Zero-Volume and Proposed Revision Plan**
> > > > > > > > > >
> > > > > > > > > > We have always been very grateful for the time and efforts the reviewer has devoted, and believed that discussions like this are crucial to improve and benefit the research community in general and should always be encouraged.
> > > > > > > > > >
> > > > > > > > > > Most of our discussions come back to the claim centered on **the zero-volume for high-dimensional sphere**, and **how can we ensure the correctness of the paper** as possible as we can, which is the mutual goal for all of us including authors and reviewers/ACs as responsible researchers.
> > > > > > > > > > Bearing this in mind, we thus summarize our discussions below.
> > > > > > > > > >
> > > > > > > > > > ---
> > > > > > > > > > - **Correctness of the manuscript/analysis/methods:** This statement initially comes from one sentence we wrote in the main manuscript in Line 180, as stand-alone and supplementary information when describing the geometric and probabilistic properties of high-dimensional spheres. It has not been used anywhere else in the paper, and we **ensure the correctness of the analytical results and the methods, as well as the strong SOTA performance despite no additional learning**.
> > > > > > > > > >
> > > > > > > > > > ---
> > > > > > > > > > - **Correctness of the statement:** This particular **zero-volume** claim is correct under certain preconditions, like every other math property. While the term itself may be not well-defined (the portion from the curse of dimensionality or actual volume from math?), we agree with the reviewer that the interpretation of the actual volume also makes a lot of sense. So if we restrict ourselves to the definition the reviewer has proposed, we then have two following situations.
> > > > > > > > > >
> > > > > > > > > >    - 1. The statement is true given the condition of the fixed radius of a high-dimensional sphere. This is what we initially wanted to convey in the paper and during our early discussions in the rebuttal, our rationale is: given pre-trained DDPMs, the expected radius is fixed in our context.
> > > > > > > > > >
> > > > > > > > > >    - 2. The statement is not true given the condition of varying radius for a high-dimensional sphere, as pointed out by the reviewer in the discussion, which describes a more generazible mathematical situation.
> > > > > > > > > >
> > > > > > > > > >    - 3. But luckily, we believe that given either condition, the statement **One needs to increase the radius of the sphere to nearly $\sqrt{d}$ before there is a non-zero concentration measure** is correct.
> > > > > > > > > >
> > > > > > > > > > ---
> > > > > > > > > > - **Proposed Solution:** Therefore, we propose to use the statement from 3. in the final paper to avoid any potential confusion may cause for future readers. Actually, we think this could **better explain and validate the actual claim we used for the actual methodology design, "adding stochasticity is a mitigation solution for the distance effect, because it brings the inverted latent encodings back to the area with higher concentration mass"**, as supported by our empirical results, now plus this theoretical clarification.
> > > > > > > > > >
> > > > > > > > > > We are looking forward to hearing what the reviewer thinks about the above plans, thank you.

---

> > > > > > > > > > > ### Comment · Reviewer_TWGA · 2023-08-18
> > > > > > > > > > >
> > > > > > > > > > > Thank you for the clarifications. First of all, I agree with the reviewer that the paper should be evaluated as a whole. This is why I did not lower the rating any further. As I mentioned in my previous comment, I think this sentence is not critical to the contribution of this paper. If this sentence were related to the main method, the result would have been different. However, I cannot evaluate this paper in the positive half because it includes mathematically incorrect statement.
> > > > > > > > > > >
> > > > > > > > > > > **Here are my suggestions for the revision:**
> > > > > > > > > > >
> > > > > > > > > > > - Let's not use the word volume. In high dimensional Gaussian distribution, "most of the probability mass is concentrated in a thin shell at large radius." (Exercise 1.20, [1]). This is a widely known fact, as can be seen from the fact that the reference is [1]. **But, the geometric volume and probability mass of this thin shell does not go to zero as $d \rightarrow \infty$. Only the width of this thin shell goes to zero.**
> > > > > > > > > > >
> > > > > > > > > > > - Based on the preceding clarification (*Correctness of the statement 1*), it appears that the author's intention is to analyze the dynamics given a predetermined high-dimensional space. If so, the expression in Line 180 is not appropriate. The sentence in Line 180 is discussing about the dynamics as $d \rightarrow \infty$. **My suggestion is that Line 178-179 is enough. Line 180 is not necessary.** **Alternatively, providing the proof or the exact reference for L180, i.e., $\lim_{d \rightarrow \infty} [V_{r}(d)-V_{r_0}(d) ] = 0$,  is enough for evaluating this paper positively.**
> > > > > > > > > > >
> > > > > > > > > > > - Honestly, I cannot understand the intention of the author in (*Correctness of the statement 2,3*). In high-dimensional Gaussian, most of the probability mass is concentrated in a thin shell at radius $r=\sigma \sqrt{d}$ (Exercise 1.20, [1]). **I am not varying the radius. If we change the dimension, the radius of the concentration shell changes.**
> > > > > > > > > > >
> > > > > > > > > > > [1] Bishop, Christopher M., and Nasser M. Nasrabadi. Pattern recognition and machine learning. Vol. 4. No. 4. New York: springer, 2006.

---

> > > > > > > > > > > > ### Author Response · Authors · 2023-08-18
> > > > > > > > > > > >
> > > > > > > > > > > > Thank you for the suggestions for the revision!
> > > > > > > > > > > >
> > > > > > > > > > > > ---
> > > > > > > > > > > > **Suggestion 1:** Sounds good, let's just not use the word "volume". We feel the "mathematically incorrect statements" may be a strong accusation, as we still have not find a formally established definition for the term in mathmatics, probably due to the ambiguity of the definition and the knowledge gap. Also, just as scientific research evolves, there is always a revisit and correction of previous statements and understanding, just as previous works believed there is no meaningful semantics in the $\epsilon$-space of diffusion models.
> > > > > > > > > > > >
> > > > > > > > > > > > ---
> > > > > > > > > > > > **Suggestion 2:** Thank you for the suggestion. We will just keep Line 178-179.
> > > > > > > > > > > >
> > > > > > > > > > > > ---
> > > > > > > > > > > > **Suggestion 3:** Our intention was that the *statement 3* could be potentially beneficial due to the fact that it gives an intuitive and straightforward explanation to our methodology design by adopting stochastic denosing process to alleviate the distance effect. Because the sampling and stochastic denoising exhibits larger radius that is closer to the shell where most probability mass locates.

---

> > > > > > > > > > > > > ### Comment · Reviewer_TWGA · 2023-08-18
> > > > > > > > > > > > >
> > > > > > > > > > > > > Thank you for the significant efforts during the rebuttal. I would raise my rating to 5.

---

> > > > > > > > > > > > > > ### Author Response · Authors · 2023-08-18
> > > > > > > > > > > > > >
> > > > > > > > > > > > > > Thank you for the in-depth questions and insightful discussions, as they are also very helpful for us as authors, to enhance and strengthen our work.

---

### Official Review · Reviewer_sSmU · 2023-06-23

**Soundness:** 3 good
**Presentation:** 2 fair
**Contribution:** 3 good
**Rating:** 6
**Confidence:** 4

**Summary:**

The authors argue they can disentangle image attributes in the *e* and *h*-level latent spaces at the discovered mixing step $t_m$. The mixing step can be determined by "the largest radius shift (e.g., 4)" based on "the mean of the squared distance of a latent sample from the origin (and following the generative trajectory) as the estimation of the Gaussian square." The semantic boundaries, editing direction vectors $\mathrm{n}$, are found by applying SVM to classify the corresponding latent encodings using the label annotations of their corresponding inputs. The authors emphasize that the boundary guidance performs in the single step at $t_m$ without any extra fine-tuned network. They show competitive results compared with the other methods using finetuning or guidance supervision, while the proposed method does not use them both.

**Strengths:**

S1. Without finetuning with an extra network nor the supervised learning techniques, they show remarkable qualitative results compared with recent other works.

S2. The method is simple yet effective. Without extra training procedures, downstream tasks may exhibit faster generations.

**Weaknesses:**

W1. The major concern of this work is clarity and reproducibility. Section 5 is the most critical part, but the authors did not allocate enough space to describe the key details of their method. Unfortunately, the background and motivation (Sec. 2-4) are also unclear due to the lack of rigorousness. And the authors should indicate the corresponding Appendix section number to refer to (I hope the authors did not circumvent the deadline in this way). Overall, the manuscript needs a major revision for the following issues (along with W2 and W3):

* W1a. In L175-176, what do you mean by "The spatial difference remains after the deterministic denoising via $p_i$ due to the homogeneity of intermediate latent spaces" in the context? The contextual understanding is laborious here. Could you elaborate on this? The paper should be self-contained and self-explainable. You should restate the definition of "homogeneity" from Asyrp [22] ("The same shift in this space results in the same attribute change in all images.") and explain or elaborate on why the homogeneity and remaining spatial differences are connected.

* W1b. In L181-182, which section in Appendix? And what is in there?

* W1c. In Fig. 2, what is "$0(\frac{r}{\sqrt{d}})$"?

* W1d. In L217-222, how do you define "Gaussian square," "squared distance," and "the radius shift"? There are ambiguities to resolve.

W2. In L149-153, it is hard to understand how you define the radius $r$. In L151, the authors said, "the radius of a high dimensional Gaussian space is defined as the square root of the expected squared distance: $r = \sigma \sqrt{d}$"; however, the text and the equation are not matched. The radius is only determined by the "expected squared distance" $d$? But, $d$ is the dimension of a sample (L131). In L152, what is "squared length"? Do you mean the squared vector length of a sample vector $\mathrm{x}$?

W3. In L228-243, "Semantic Boundary Search" omitted major details. How do you select the "semantic separation hyperplane" to the corresponding annotations, e.g., "smile"? The authors failed to relate and indicate to refer to Appendix E. In Appendix E, Theorem E.1. has a typo $\|$ or $|$ and a missing proof for the Theorem. Even though consulting with Appendix, the reproducibility seems to be infeasible.

**Questions:**

Q1. In L70, omitted a space after "segmentation consistency."

Q2. In L105, normal text for real images $x_0$ instead of a vector representation $\mathrm{x}_0$ for consistency. In L120, $x$ is the latent encodings. The inconsistency makes me confused.

Q3. In L136, "the stochastic DDPMs and the deterministic DDIMs" may link to Appendix A.2. But the authors rarely refer to their corresponding Appendix sections. For DDIMs, $\eta$ controls the degree of stochasticity. Why do you call "deterministic DDIMs"? Did you set $\eta=0$?

Q4. In L297, "The quantitative evaluations are shown in (Table) 3", missing "Table"?

**Limitations:**

None to mention.

---

> ### Author Rebuttal · Authors · 2023-08-05
>
> We have submitted our **anonymized code to AC to eliminate the reproducibility concerns** of the reviewer, and ensure that the reported results and performance are easily reproducible. We are also thankful for the careful reading and helpful suggestions on the writing, such as indicating the specific appendix sections in the main paper, which we would like to revise accordingly in the final version. We note that the reviewer is concerned with the ambiguous definitions of multiple metrics and concepts in high-dimensional Gaussian, we would like to politely clarify that all the concepts and calculations of adopted metrics such as Gaussian radius are **well established in mathematics [2]**, which we utilize as tools to better analyze and understand DMs, but **didn’t define or invent those by ourselves**.
>
> ---
> - **Q1:** In L175-176, what do you mean by "The spatial difference remains after the deterministic denoising via due to the homogeneity of intermediate latent spaces" in the context? Explain or elaborate on why the homogeneity and remaining spatial differences are connected.
>
> - **A1:** This echoes the statement “The same shift in this space results in the same attribute change in all images” from [1]. In our context, the homogeneity means that spatial difference in terms of the Gaussian radius (i.e., inverted latent encodings have a smaller radius than sampled ones) along the trajectory, as shown in Tab. 2 in the main paper and Tab. 1 and 2 in the appendix.
>
>    The connection between the semantic homogeneity and the homogeneous spatial difference is due to the intrinsic nature of semantics formed during the training. In other words, the semantic boundaries in $\epsilon$-space **remain invariant** after its formation in training at our introduced mixing step, which in return makes our BoundaryDiffusion method possible with **one single step** operation. Otherwise, if the homogeneity of semantics and the spatial boundaries vary, we would not be able to preserve the editing effect on target attributes with one-step editing.
>
> ---
> - **Q2:** In L181-182, which section in Appendix? And what is in there?
>
> - **A2:** In Appendix B High Dimensional Space, we provide necessary theoretical foundations for understanding the geometric and probabilistic properties of high-dimensional Gaussian. In particular, we show the **theoretical support for the thin annulus pattern of concentration mass illustrated in Fig. 2**, and demonstrate the statistical significance of the spatial difference computed in Tab.1 and 2.
>
> ---
> - **Q3:** In Fig. 2, what is $O(\frac{r}{\sqrt{d}})$?
>
> - **A3:** This is **the width of the slice** for the concentration mass, as stated in L178-179, “ the probabilistic concentration mass of a high-dimensional sphere is located within a thin slice at the equator”. This is an established property of high-dimensional Gaussian, with theoretical support given in Lemma B.4 in appendix.
>
> ---
> - **Q4 & Q5:** In L217-222, how do you define "Gaussian square", "squared distance," and "the radius shift"? There are ambiguities to resolve. In L149-153, it is hard to understand how you define the radius $r$.
>
> - **A4 & A5:** We didn’t define the above concepts by ourselves, they are **established metrics and concepts** in mathematics, as in the book [2]. Hope the explanation below will give a better definition of these metrics, and we are glad to include the clarification in our final version.
>
>    *Consider a $d$-dimensional Gaussian centered at the origin with variance $\sigma^2$. For a point $\bf{x} = (x_1, x_2, ..., x_d)$ chosen at random from the Gaussian, the expected squared length of $\bf x$ is: $E(x^2_1+x^2_2+...+x^2_d)=dE(x^2_1)=d\sigma^2$. For large $d$, the value of the squared length of $\bf x$ is tightly concentrated about its mean. The square root of the expected squared distance (namely $\sigma\sqrt{d}$) is called Gaussian radius.*
>
> ---
> - **Q6:** In L228-243, how do you select the "semantic separation hyperplane" to the corresponding annotations, e.g., "smile".
>
> - **A6:** As stated in L230, we use an SVM linear classifier to fit the classification boundary using a few annotated images. For example, we prepare 100 human face images with 50 annotated as smile and 50 as non-smile. We fit an SVM to classify those samples, then the parameters in SVM are the hyperplanes. We can further use those parameters to find the unit normal vector $\bf n$ and use it as the directional guidance in Eqn. 4.
>
> ---
> - **Q7:** In L136, "the stochastic DDPMs and the deterministic DDIMs". For DDIMs, $\eta$ controls the degree of stochasticity. Why do you call "deterministic DDIMs"? Did you set $\eta=0$?
>
> - **A7:** Yes, $\eta$ controls the stochasticity, we use $\eta=0$ for deterministic DDIMs.
>
> ---
> - **Other suggestions on format including space and notations.**
>
> - We appreciate the suggestions and will revise accordingly.
>
> ---
> [1] Diffusion models already have a semantic latent space. In ICLR, 2023.
>
> [2] Foundations of data science. Cambridge University Press, 2020.

---

> > ### Comment · Reviewer_sSmU · 2023-08-11
> > **Still not sure about the improvement of presentation for publication**
> >
> > I requested the definitions of concepts in the work for the *self-contained* presentation for readability. Although the soundness and contribution pass the bar in my previous evaluation, I still have concerns about your presentation (+ more sure about soundness). To reassess the score, I need a more specific revision plan to improve the readability and clarity of writing, reflecting your rebuttal and below additional feedback.
> >
> > A1. How do you revise the manuscript according to your explanation above?
> >
> > A2. It would be thankful to refer to the corresponding Appendix across the manuscript faithfully. Could you summarize the editing plan for this matter?
> >
> > A3. It seems you used the big-O notation; however, the figure is ambiguous since you didn't state it correctly, and hard to infer by using a non-conventional font family.
> >
> > A4&5. Thank you for the kind explanation. For the self-contained description, do you plan to incorporate (restate, of course) the definition of Gaussian radius and others with proper citation in the manuscript?
> >
> > A6. You could add those explanations in the manuscript to better understand what you did.
> >
> > A7. Thank you for the clarification.

---

> > > ### Author Response · Authors · 2023-08-11
> > > **Concrete Revision Plans**
> > >
> > > Thank you for reading our rebuttal and for the further suggestions! Yes, we agree that a **self-contained and clear** presentation would help the readers to better understand the merits of our work. Following the suggestions, we propose the requested revision plan as below.
> > >
> > > ---
> > > - **1.** How do you revise the manuscript according to your explanation above?
> > >
> > > - **Revision 1:** We will add the **explicit homogeneity explanation** (*The same shift in this space results in the same attribute change in all images*) from [1] in our main paper, and **discuss the connections** as we have replied in the rebuttal to provide a more comprehensive and intuitive explanation.
> > >
> > > ---
> > > - **2.** It would be thankful to refer to the corresponding Appendix across the manuscript faithfully. Could you summarize the editing plan for this matter?
> > >
> > > - **Revision 2:** Thank you for the suggestion. We will **revisit and revise the entire paper** and ensure that every time when we refer to the appendix in the main paper, the **specific section number is included**.
> > >
> > > ---
> > > - **3.** It seems you used the big-O notation; however, the figure is ambiguous since you didn't state it correctly, and hard to infer by using a non-conventional font family.
> > >
> > > - **Revision 3:** Thank you for pointing out the font. We will explicitly explain the big-O notation in the **figure caption**, and **adjust the font** in the figure accordingly.
> > >
> > > ---
> > > - **4&5.** For the self-contained description, do you plan to incorporate (restate, of course) the definition of Gaussian radius and others with the proper citation in the manuscript?
> > >
> > > - **Revision 4&5:** For sure! We promise to **explicitly restate the definitions with proper citations** to avoid potential confusion for readers.
> > >
> > > ---
> > > - **6.** You could add those explanations in the manuscript to better understand what you did.
> > >
> > > - **Revision 6:** Yes, we will add the explanations as suggested in our final version.
> > >
> > > ---
> > > Please let us know if you have any further concerns!

---

> > > > ### Comment · Reviewer_sSmU · 2023-08-14
> > > > **Implementation details of Table 1**
> > > >
> > > > Sorry for not bringing in the first review; I found it after rereading the manuscript for assessment.
> > > >
> > > > How were the confidence intervals of the numbers in Table 1? Since you didn't mention the number of samples $N$ and the variances, we cannot evaluate whether the iDDPM significantly decreases $\Delta r$.

---

> > > > > ### Author Response · Authors · 2023-08-14
> > > > > **Responses to the implementation details of Table 1**
> > > > >
> > > > > Thank you for the question!
> > > > >
> > > > > In the implementation for calculating the results of Table 1, we initially used 1000 random samples (either sampled from Gaussian distribution or inverted from real images) to calculate the Gaussian radius. Later during our continual experiments, we found **the results very consistent** given even fewer samples (~200).
> > > > >
> > > > > To answer this particular raised question: *whether the iDDPM significantly decrease $\Delta r$ ?* We quickly re-ran 3 times the iDDPM-AFHQ-256 model for the inverted setting with 200 randomly selected real images as further confirmation, and get the results of an average of 437.98 with a variance 0.05. Therefore, we are **confident that the reported numbers are valid and iDDPM indeed decreases the $\Delta r$** quite significantly compared to the DDPM.
> > > > >
> > > > > We will add this implementation detail to our appendix section G, and incorporate the variance to all the reported results in Table 1 in the main manuscript.

---

> > > > > > ### Comment · Reviewer_sSmU · 2023-08-16
> > > > > >
> > > > > > Very nice to hear that. Upon the authors' faithful feedback and detailed revision plans, I raised my score toward acceptance.

---

### Official Review · Reviewer_EhMb · 2023-07-03

**Soundness:** 3 good
**Presentation:** 3 good
**Contribution:** 3 good
**Rating:** 7
**Confidence:** 3

**Summary:**

This paper proposes a method for semantic control in image editing using pre-trained denoising diffusion models (DDMs) without the need for additional network modules. The method involves optimizing the denoising trajectory solely via frozen DDMs and exploring the critical step in the denoising trajectory that characterizes the convergence of a pre-trained DDM. The paper also presents a method to search for the semantic subspaces boundaries for controllable manipulation by guiding the denoising trajectory toward the targeted boundary at the critical convergent step. The paper leverages ideas in high-dimensional statistics, estimating the Gaussian radius and studying the difference between DDPM sampling and DDIM inversion. The method can perform semantic manipulation in one step and is extensively validated through experiments and datasets.

**Strengths:**

* The paper provides a comprehensive analysis of the latent spaces and the trajectory for pre-trained DDMs from a high-dimensional space perspective. This analysis can help better understand the behavior and properties of these models. Particularly interesting is the analysis of DDPM sampling vs DDIM inversion.

* The authors propose a method to achieve semantic control and editing by guiding the denoising trajectory using the semantic boundaries in high-dimensional latent spaces with pre-trained and frozen DDMs. This approach leverages the stochasticity and the mixing step for better qualitative results and more effective manipulation. The method is grounded in theory and effective in practice.

* The paper explores the potential of pre-trained diffusion models without learning any additional network model modules. This approach can be more efficient and less complex than methods that require additional learning.

**Weaknesses:**

* In this paper, certain assumptions are made about the structure of the latent space which are difficult to verify or justify. Hierarchical latent spaces are generally difficult to comprehensively analyze and validate due to their non-obvious, often high-dimensional, characteristics.

* Furthermore, the terminology used for the methods, namely 'semantic boundary search' and 'trajectory mixing', creates an additional layer of complexity and is confusing. In particular trajectory mixing might be misunderstood or misinterpreted. You claim to perform semantic control in a single step but then your main method is called trajectory mixing. I would simplify or clarify the notation.

**Questions:**

* Regarding Equations 4 and 5, could you provide a more detailed explanation? Although the introduction is well-articulated, the presentation of these methods seems somewhat brief and rapid. Could you break them down further for better understanding?

* As for Figure 3, could you specify what exactly you are 'mixing'? Furthermore, could you clarify the term 'mixing' within this context? What is its precise meaning in this scenario?

**Limitations:**

* Extensive experiments and grounded method. I do not see any relevant limitation to this method that can be applied to any pretrained DDPM.

---

> ### Author Rebuttal · Authors · 2023-08-05
>
> We deeply appreciate the acknowledgement and the insightful questions, which we would like to include the clarifications in our final paper.
>
> ---
> - **Comment 1:** In this paper, certain assumptions are made about the structure of the latent space which are difficult to verify or justify. Hierarchical latent spaces are generally difficult to comprehensively analyze and validate due to their non-obvious, often high-dimensional, characteristics.
>
> - **Response to Comment 1:**
>
>    While we believe the analysis from the novel perspective of high-dimensional latent space helps to build a more comprehensive and better understanding of diffusion models, we also agree with the reviewer that arbitrary high-dimensional distributions usually exhibit very complex spatial behaviors and characteristics in mathematics and statistics.
>
>    Luckily, Gaussian has many well-established properties that help us to better interpret DMs, which also motivates us to consider the Gaussian prior as a **reference** to understand DMs, and restrict our analytical studies mostly to the latent space of step $T$, which exhibits a **standard Gaussian by definition**. We are **being very prudent to make assumptions and in the choice of mathematical tools**. For instance, when we propose the automatic research method for the mixing step using the Gaussian radius as the metric, we indeed notice and explicitly state that the intermediate latent spaces do not exhibit standard Gaussian as the trajectory goes more toward real data space (see details in Appendix D.2), but the Gaussian radius remains to be a reasonable approximation to trace the latent spatial behaviors (turns out to be also pretty good as demonstrated in our empirical experiments in various downstream tasks by achieving strong SOTA performance) before the mixing step.
>
>    We do not boastfully claim to fully understand the DMs in this work, but we are also confident that it should help to build a better and more comprehensive understanding of diffusion generative models for interested researchers in the community, with abundant theoretical and empirical evidence.
>
> ---
> - **Q1:** Regarding Equations 4 and 5, could you provide a more detailed explanation? Although the introduction is well-articulated, the presentation of these methods seems somewhat brief and rapid. Could you break them down further for better understanding?
>
> - **A1:** For sure. The overall idea of our *BoundaryDiffusion* consists of:
>
>    **i)** Locate and fit the semantic boundaries (characterized by their unit normal vector $\bf n$ at the mixing step $t_m$ via SVM using a few labeled samples (the parameters from SVM characterize the boundary itself, the unit normal vector can be directly computed using zero dot-product). Note this is for locating the semantic boundaries that *formed and stayed invariant during the training* of DMs.
>
>    **ii)** Impose boundary-guided editing signals $d$ to a latent encoding $x_{t_m}$ at mixing step. Note the latent encoding can come from different sources at step $T$: for conditional semantic manipulation, it comes usually from an inverted image; for unconditional semantic control, it comes from a random sample of Gaussian. The denoising trajectory from $T$ to $t_m$ is deterministic following DDIM.
>
>    **iii)** Follow the remaining stochastic denoising trajectory to step $0$.
>
>    **For Eqn. 4**, it defines the editing signal $d$ with respect to an identified semantic boundary characterized by its unit normal vector $\bf n$.
>
>    **For Eqn. 5**, it defines the entire mixing trajectory process with the mixing step $t_m$ as a separation point. Before $t_m$, the sampling/generation process follows a deterministic denoising trajectory via DDIM [1]. After $t_m$, the denoising trajectory is stochastic via the original DDPM [2].
>
> ---
> - **Q2:** As for Figure 3, could you specify what exactly you are *mixing*? Furthermore, could you clarify the term *mixing* within this context? What is its precise meaning in this scenario?
>
> - **A2:**  Thank you for the question. There are two main reasons for us to use the term *mixing* within the context of *BoundaryDiffusion* in Fig. 3. **Firstly**, the editing operation we proposed is a one-step modification on the latent encoding at the **mixing step $t_m$**, so we find it appropriate to introduce the idea of the mixing step in the methodology design and naming. **Secondly**, we use two denoising techniques, namely the deterministic and stochastic, along the trajectory, which is also a **mixture** that relates to the idea of *mixing*.
>
> ---
> [1] Denoising Diffusion Implicit Models. In ICLR, 2021.
>
> [2] Denoising diffusion probabilistic models. In NeurIPS 2020.

---

> > ### Comment · Reviewer_EhMb · 2023-08-11
> >
> > Thank you for the clarifications. I read the rebuttal and my questions have been answered. I still think that "trajectory mixing" is not a great name for your method.

---

> > > ### Author Response · Authors · 2023-08-11
> > >
> > > Thank you for reading our rebuttal, and glad that we have answered your questions!
> > >
> > > Following the suggestion, we will rename the method just as *BoundaryDiffusion* without mentioning the mixing trajectory to avoid potential confusion, and change the title accordingly if this is allowed.

---

### Official Review · Reviewer_5XEa · 2023-07-06

**Soundness:** 2 fair
**Presentation:** 3 good
**Contribution:** 2 fair
**Rating:** 5
**Confidence:** 3

**Summary:**

This paper proposes Boundary Diffusion, a method for controlling the output of frozen pre-trained diffusion models. Boundary Diffusion contains two parts: (1) an analysis from the characteristics of Gaussian distribution to determine the key mixing step (2) a boundary guidance method to control the output of the diffusion models. Empirical results are presented to showcase the effectiveness of the proposed method.

**Strengths:**

1. The paper is well-organized and the contents are easy to digest.

2. The analysis based on the characteristics of Gaussian distribution is an interesting angle.

3. The empirical results in Figure 1 is intriguing.

**Weaknesses:**

**1. Missing important baselines**: Since in the proposed method needs to train an SVM with noisy data x_{tm}, classifier guidance should be compared as an important baseline. Classifier guidance requires the user to train extra classifiers upon all the noisy data over all the time steps, which makes it more expensive than the proposed method, but the computational cost of SVMs are small in the deep learning era and I think it should be compared due to the great similarity with the proposed method. The computational cost of classifier guidance can be further reduced by applying it on a fraction of time steps (instead of all the 1000 steps). Anyway, it should not be ignored.

**2. Need additional experiments**: the quantitative results are only available for CelebA+smile in Table 3. This makes readers doubt the generalization ability of the proposed method. Because the emphasis of this paper is empirical contribution rather than theoretical analysis, the authors should provide more quantitative results and comparison on different models, datasets and semantic attributes. For example, the LSUN-Church models.


If the authors could provide the required experimental results, I am happy to increase my ratings.

Typos: L222, Tab 3 ==> Tab 2.

Additionally, the following recent work is related to this paper and the authors may be willing to discuss it as a concurrent work:

FlowGrad: Controlling the Output of Generative ODEs With Gradients (https://openaccess.thecvf.com/content/CVPR2023/papers/Liu_FlowGrad_Controlling_the_Output_of_Generative_ODEs_With_Gradients_CVPR_2023_paper.pdf)

**Questions:**

N/A

**Limitations:**

Limitations and broader impacts are properly discussed.

---

> ### Author Rebuttal · Authors · 2023-08-06
>
> We are very grateful for the valuable comments, especially with the suggestion to include more experimental results to further enhance our paper.
>
> ---
> - **Comment 1:** Baseline comparison with the classifier guidance.
>
> - **Response to Comment 1:**
>
>    We appreciate the comment. As suggested, we implement the CG following the original code [2], but with the **same base models trained on CelebA-HQ-256** to ensure a fair comparison, and report the quantitative results in Tab.1 below.
>
>    In terms of the computational cost, Classifier guidance [2] learns the gradients along the trajectory at each intermediate step to achieve the guided control. Conceptually it is similar to Asyrp [3] which modifies the intermediate latent variables via auxiliary neural networks. In our rebuttal, we train the classifier for approximately 30 mins with CG, similar to DiffusionCLIP [3] and Asyrp[4]. It is worth noting that our *BoundaryDiffusion* **does not train classifiers**, but rather **search** and **fit** the existing boundaries with SVM as hyperplanes in **negligible time about 1s**, which is also one of the key differences that distinguish our work from other learning-based methods since we are exploring the **intrinsic semantic properties** of pre-trained DMs formed in training. We believe this high-level take-away is critical since we proved that for a probabilistic generative model trained without semantic supervision, it forms meaning semantics by itself. On the contrary, researchers used to think that unconditionally trained generative models are not good at learning data semantics.
>
> | Method             | $S_{dir}$ $\uparrow$     | SC $\uparrow$        | ID $\uparrow$       | FID  $\downarrow$      |
> |--------------------|----------|-----------|----------|-----------|
> | ClassifierGuidance [2] | 0.15     | 86.5%     | 0.65     | 70.71     |
> | DiffusionCLIP [3]      | 0.17     | **93.7%** | 0.70     | 86.23     |
> | Asyrp [4]              | **0.19** | 87.9%     | -        | 68.38     |
> | FlowGrad+RF [1]       | -        | -         | **0.74** | -         |
> | *Our BoundaryDiffusion*  | 0.17     | 90.4%     | 0.73     | **68.14** |
> Tab. 1 More baseline comparisons with additional baselines. Note among all the comparing methods, *BoundaryDiffusion* is the only learning-free method with **one-step** guidance operation.
>
> ---
> - **Comment 2:** Need additional quantitative results. For example, LSUN-Church dataset.
>
> - **Response to Comment 2:**
>
>    We are happy to provide more quantitative results to demonstrate the effectiveness and SOTA performance of our method. As suggested, we report additional results on more datasets with different attributes. Note that the face identity similarity metric is no longer applicable to datasets other than human faces, as shown in Tab. 2, 3, and 4. In particular, we note our *BoundaryDiffusion* achieves higher segmentation consistency (SC) for LSUN-Church, partially due to the reason that we are modifying the latent encoding with **minimum shift** in latent space to cross the boundary, thus better preserves the original structure compared to learning based SOTA. We also included **more non-cherry-picky** qualitative examples for LSUN-Church in our one-page rebuttal for your reference, to ensure the effectiveness and SOTA performance as claimed.
>
> | Method            | $S_{dir}$  $\uparrow$   | SC $\uparrow$    | ID $\uparrow$      |
> |-------------------|----------|-----------|----------|
> | DiffusionCLIP [3]    | 0.17     | **93.1%** | 0.63     |
> | Asyrp   [4]          | **0.18** | 88.3%     | 0.67     |
> | *Our BoundaryDiffusion* | **0.18** | 92.5%     | **0.69** |
> Tab. 2 Results on CelebA-HQ-256, with attribute “add or remove glass”.
>
>
> | Method            | $S_{dir}$ $\uparrow$    | SC  $\uparrow$     |
> |-------------------|----------|------------|
> | DiffusionCLIP  [3]   | 0.96     | 65.02%     |
> | Asyrp          [4]   | **0.98** | 65.83%     |
> | *Our BoundaryDiffusion* | 0.97     | **66.01%** |
> Tab. 3 Results on LSUN-Church-256, with example attribute “red brick church”.
>
> | Method            | $S_{dir}$  $\uparrow$   | SC  $\uparrow$       |
> |-------------------|----------|------------|
> | DiffusionCLIP [3]    | 0.91     | 64.82% |
> | Asyrp  [4]           | **0.94** | 62.65%     |
> | *Our BoundaryDiffusion* | **0.94** | **65.17%**     |
> Tab. 4 Results on LSUN-Church-256, with example attribute “ancient church”.
>
> ---
> - **Others:** [1]  is related to this paper and the authors may be willing to discuss it as a concurrent work.
>
> - **Response:** Thank you for the suggestion! We have read the FlowGrad paper and agree that it is a very interesting work related to ours and would like to discuss it in our final version! FlowGrad tackles the ODE-based generative models and proposes to control the generated output by controlling the back-propagated gradient to intermediate time steps on the ODE trajectory.
>
>    Essentially, FlowGrad features a method that optimizes the editing gradient along the ODE trajectory of pre-trained models. In that sense, we are similar in a way to seek control of the generative models in a more disentangled way. While FlowGrad proposed to do so via decomposing the back propagation process, we seek to track the gradual evolution of high-dimensional latent space along the trajectory.
>
> ---
> [1] FlowGrad: Controlling the Output of Generative ODEs with Gradients. In CVPR 2023.
>
> [2] Diffusion Models Beat GANs on Image Synthesis. In NeurIPS 2021.
>
> [3] Diffusionclip: Text-guided diffusion models for robust image manipulation." In CVPR 2022.
>
> [4] Diffusion models already have a semantic latent space. In ICLR 2023.

---

> > ### Comment · Reviewer_5XEa · 2023-08-14
> > **Thank you**
> >
> > I think the rebuttal addresses my concerns. I encourage the authors to include the additional results into the later versions. I will increase my rating to a 5.

---

### Author Rebuttal · Authors · 2023-08-06

### Overall responses to all reviewers and ACs

We thank all reviewers and ACs for the valuable feedback and for initiating insightful discussions. We also appreciate the acknowledgment of our work from both theoretical and experimental aspects, with the *theoretical analysis to better understand diffusion models from a novel perspective* (R-5XEa, R-EhMb, R-TWGA), and *strong experimental performance even without additional network learning* (R-5XEa, R-sSmU). (Note our **non-cherry-picky** results show better fidelity and more natural editing effects than existing learning-based SOTA methods due to minimum and effective shift in latent spaces.)

From our perspective as authors, our **key take-away/biggest contribution** in this work is: **we proved that an unconditionally trained generative diffusion model can well capture the data semantics in a unified latent space along the denoising trajectory and form invariant and separable boundaries at mixing steps.**

This is **not evident** in previous literature where probabilistic generative models, including DMs, are conventionally believed to be not good at learning data semantics in unconditional settings **without explicit semantic supervisions** such as text (see examples with multiple popular text-to-image generation DMs). We demonstrate the above **strong but important claim** with both theoretical and empirical support via proposed analysis from the high-dimensional angle and *BoundaryDiffusion* method and believe this should greatly benefit our community with a better understanding of diffusion models. In addition, this should also help to **bridge the gap between generative and discriminative models**, with respect to their different emphasis to **learn distributions and discriminative decision boundaries**, and shall benefit even **discriminative downstream tasks** such as segmentation [1], which uses pre-trained DMs as representation learning models to extract semantic features.

We are also confident in the SOTA performance claim and have submitted our anonymized code to AC in a separate comment during rebuttal, and would like to invite any interested reviewers/AC to check and test the code to ensure reproducibility.

We note different reviewers are intrigued with either theoretical or empirical aspects of our work, but we found both to be indispensable in proving the above claim and inspiring our methodological design of a **simple, elegant, effective yet non-intuitive** framework.
To restate the contributions and novelties of our work, we make the table below as further clarifications.

|   Work  |        Study on the latent space     |         Study on mixing step   |    Methodology logic                           |      Applications    |      Auxiliary models     |     Trainable param.   |      Training time     |
|---------|----------------------------------------|-----------------------------|-----------------------|--------------------|--------------------|------------------------|-----------------|
| DiffAutoencoder [CVPR22 oral] |                No semantic meaning                      |                                                         Non                                                         |              Learn an attached autoencoders on $x$-level             |           Unconditional synthesis          |          MLP based autoencoder          |   Several million based on architecture   |                    ~28 GPU hours            |
|        DiffusionCLIP [CVPR22]       |                    Non                   |                Pure empirical, manual selection based on final output                             | Fine-tune the entire model on $x$-level, operates on iterative steps |            Guided image editing            | No, but fine-tune the pre-trained model | Same as base model, thus several millions |             ~20-30min for each attribute            |
|      Asyrp [ICLR2023 spotlight]     |                     Already have a semantic space, but only on $h$-level                     |                              Pure empirical, manual selection based on final output                             |                Learn editing signal on iterative steps               |            Guided image editing            |        Auxiliary editing networks       |                   ~800k                   |             ~20-30min for each attribute            |
|                Ours      | **Both $h$ and $x$ levels have semantic space**, can be well leveraged to control via our method | **Theoretical interpolation and deviation, automatic search via Gaussian radius  (thus not dependant on final output)** |             Mixing step boundary guidance,  **single step operation**             | **Unconditional synthesis + Guided image editing** |                    **No**                   |                    **0** (we are fitting existing hyperplanes instead of learning classifiers)                    | **Negligible, all attributes at the same latent space** |

---
[1] Open-Vocabulary Panoptic Segmentation with Text-to-Image Diffusion Models, In CVPR 2023.

---

> ### Author Response · Authors · 2023-08-14
> **Final responses from authors**
>
> We sincerely appreciate the efforts from all reviewers, for their insightful questions during the rebuttal and their acknowledgment.
>
> We have confirmed in our follow-up additional experiments that the proposed *BoundaryDiffusion* is also **applicable to LDMs (StableDiffusion)**, meaning that the latent image representation from encoder-decoder also preserves the invariant semantics we have proposed in this work for text conditioned diffusion models. We will release the codes and integrate semantic boundaries we have located into the public platform (e.g., HuggingFace) to allow easy access for the research community.
>
> We will also incoprate the revisions promised during the rebuttal in our final paper to enhance this work. Thank you all!

---

### Decision · Program_Chairs · 2023-09-21

**Decision:**

Accept (poster)

**Comment:**

The authors have put significant effort into their rebuttal and there were intensive discussions between the reviewers. In the end, all critical reviewer concerns have been addressed and the reviewers now unanimously vote for acceptance.
-> accept. Congratulations!
When preparing your camera-ready, please take all reviewer comments into account as promised in the rebuttal.